# The malaria parasite sheddase SUB2 governs host red blood cell membrane sealing at invasion

Christine R Collins[1†], Fiona Hackett[1†], Steven A Howell[2], Ambrosius P Snijders[2], Matthew RG Russell[3], Lucy M Collinson[3], Michael J Blackman[1,4]*

[1]Malaria Biochemistry Laboratory, The Francis Crick Institute, London, United Kingdom; [2]Protein Analysis and Proteomics Platform, The Francis Crick Institute, London, United Kingdom; [3]Electron Microscopy Science Technology Platform, The Francis Crick Institute, London, United Kingdom; [4]Faculty of Infectious Diseases, London School of Hygiene & Tropical Medicine, London, United Kingdom

**Abstract** Red blood cell (RBC) invasion by malaria merozoites involves formation of a parasitophorous vacuole into which the parasite moves. The vacuole membrane seals and pinches off behind the parasite through an unknown mechanism, enclosing the parasite within the RBC. During invasion, several parasite surface proteins are shed by a membrane-bound protease called SUB2. Here we show that genetic depletion of SUB2 abolishes shedding of a range of parasite proteins, identifying previously unrecognized SUB2 substrates. Interaction of SUB2-null merozoites with RBCs leads to either abortive invasion with rapid RBC lysis, or successful entry but developmental arrest. Selective failure to shed the most abundant SUB2 substrate, MSP1, reduces intracellular replication, whilst conditional ablation of the substrate AMA1 produces host RBC lysis. We conclude that SUB2 activity is critical for host RBC membrane sealing following parasite internalisation and for correct functioning of merozoite surface proteins.

*For correspondence:
Mike.Blackman@crick.ac.uk

[†]These authors contributed equally to this work

Competing interests: The authors declare that no competing interests exist.

## Introduction

The phylum Apicomplexa comprises a diverse group of protozoan organisms, many of which are obligate intracellular parasites of clinical or veterinary importance. A feature of these parasites is their possession of invasive forms that actively penetrate host cells. In the asexual blood stages of infection by malaria parasites (*Plasmodium* species), merozoites invade red blood cells (RBCs), replicating intracellularly to produce merozoites that egress to invade fresh RBCs. Invasion is a rapid, multi-step process that includes binding, merozoite reorientation, discharge of secretory organelles called rhoptries and micronemes, formation of an electron-dense 'tight junction' (TJ) between the merozoite apical end and the RBC membrane, and actinomyosin-powered entry into the RBC through this structure, with concurrent formation of a membrane-bound parasitophorous vacuole (PV) within which the parasite develops (*Aikawa et al., 1978*; *Dvorak et al., 1975*; *Weiss et al., 2015*; *Bannister et al., 1975*). Invasion ends with sealing of the RBC behind the intracellular parasite, concomitant with pinching off of the nascent PV membrane (PVM) in a membrane scission event such that the PVM is eventually non-contiguous with and internal to the RBC membrane. Invasion is typically followed by transformation of the host RBC into a shrunken 'spiky' state called echinocytosis, which typically resolves within minutes. The parasite rapidly transforms into an amoeboid 'ring' form. Whilst many advances have been made over recent decades in understanding invasion, particularly in the most lethal malaria species *Plasmodium falciparum* and *Plasmodium knowlesi*, as well as in the related apicomplexan parasite *Toxoplasma gondii*, the mechanisms underlying PV formation

**eLife digest** Malaria kills or disables hundreds of millions of people across the world, especially in developing economies. The most severe form of the disease is caused by *Plasmodium falciparum*, a single-cell parasite which, once inside a human host, forces its way into red blood cells to feed on a protein called haemoglobin. This invasion relies on *P. falciparum* being engulfed by the membrane of the red blood cell, which then seals off to form a compartment inside the cell where the parasite can feed and multiply. Invasion takes less than 30 seconds, and it involves *P. falciparum* losing the coat of proteins that covers its surface. An enzyme calls SUB2 cleaves or cuts off these proteins, but exactly why and how the shedding takes place during infection is still unclear.

To investigate, Collins, Hackett et al. deactivated the gene which codes for SUB2, and examined how mutant *P. falciparum* would survive and multiply. Without the enzyme, the parasites failed to shed many of their proteins, including some that were not previously known to be removed by SUB2. The majority of the genetically modified parasites also failed to invade red blood cells. In particular, most of the host cells ruptured, suggesting that the protein coat needs to be discarded for the engulfing process to be completed properly. When the enzyme-free mutants did manage to make their way into a red blood cell, they starved to death because they could not digest haemoglobin. SUB2 and surface coat shedding therefore appears to be essential for the parasite to survive.

*P. falciparum* is fast becoming resistant to the many drugs that exist to fight malaria. New treatments that target SUB2 may therefore help in combatting this deadly disease.

are obscure and nothing is known of the molecular mechanism(s) responsible for host cell membrane sealing.

Early electron microscopic (EM) studies showed that invading merozoites shed a 'fuzzy coat' of ~20 nm-long bristle-like fibres (*Aikawa et al., 1978*; *Bannister et al., 1975*). These are likely composed predominantly of the major glycosyl phosphatidylinositol (GPI)-anchored surface protein MSP1, which is synthesised as an abundant, ~200 kDa precursor at the plasma membrane of developing parasites (*Holder et al., 1992*). Minutes before egress, MSP1 is proteolytically cleaved into several fragments by a parasite protease called SUB1 (*Koussis et al., 2009*; *Yeoh et al., 2007*). These fragments form a noncovalent complex on the merozoite surface (*McBride and Heidrich, 1987*) where, together with several partner proteins, it facilitates membrane rupture at egress (*Das et al., 2015*). During invasion the bulk of the MSP1 complex is shed from the merozoite (*Riglar et al., 2011*; *Blackman et al., 1996*; *Blackman et al., 1990*; *Blackman et al., 1991*) as a result of a single further cleavage catalysed by a second, membrane-bound protease called SUB2 (*Harris et al., 2005*; *Hackett et al., 1999*; *Barale et al., 1999*). SUB2 also mediates shedding of two merozoite surface integral membrane proteins called PTRAMP and AMA1, which are released from micronemes at around egress (*Howell et al., 2003*; *Green et al., 2006*; *Siddiqui et al., 2013*; *Thompson et al., 2004*). PTRAMP acts as an RBC binding ligand (*Siddiqui et al., 2013*) whilst AMA1 plays a central role in TJ formation (reviewed in *Harvey et al., 2014*). To access its substrates, SUB2 is also released from micronemes onto the merozoite surface, where it translocates to the posterior pole of the parasite just before or during invasion (*Riglar et al., 2011*; *Harris et al., 2005*). Despite evidence that SUB2 is essential (*Uzureau et al., 2004*; *Zhang et al., 2018*; *Bushell et al., 2017*) and that shedding of MSP1 and AMA1 is important and can aid evasion of invasion-inhibitory antibodies (*Blackman et al., 1990*; *Olivieri et al., 2011*; *Guevara Patiño et al., 1997*; *Lazarou et al., 2009*), the molecular function of SUB2-mediated shedding is unknown.

Here, we used genetic modification of *P. falciparum* SUB2 and two of its substrates to examine the essentiality and function of SUB2 in the erythrocytic lifecycle. We show that SUB2 depletion results in defects in merozoite surface protein shedding and sealing of the host RBC upon invasion, leading to either abortive invasion with loss of host RBC haemoglobin, or developmental arrest of the intracellular parasite. Our findings highlight SUB2 as a key mediator of parasite viability and host RBC membrane integrity.

## Results

### Efficient conditional ablation of SUB2 expression

The *P. falciparum sub2* gene encodes a type I integral membrane protein with a large ectodomain incorporating a subtilisin-like protease module (*Hackett et al., 1999*; *Barale et al., 1999*). We designed a construct to integrate into the *sub2* locus by homologous recombination, producing a modified locus in which the entire second exon encoding the crucial catalytic Ser961, transmembrane domain (TMD) and cytoplasmic domain was flanked (floxed) by *loxP* sites (*Figure 1A*). Integration also fused a triple hemagglutinin (HA3) epitope tag to the SUB2 C terminus, as achieved previously (*Riglar et al., 2011*; *Harris et al., 2005*). Transfection of the construct into the DiCre-expressing *P. falciparum* 1G5DC clone (*Collins et al., 2013a*) resulted in outgrowth of WR99210-resistant parasites expressing HA3-tagged SUB2. Immunofluorescence analysis (IFA) of two integrant clones of the modified parasite line (called SUB2HA3:loxP) with anti-HA3 antibodies showed a signal consistent with the previously-determined location of SUB2 in micronemes, confirming correct gene modification (*Figure 1B*).

Excision of the floxed sequence was predicted to produce a truncated gene product lacking a functional catalytic domain, TMD, and HA3 tag. To assess the efficiency of DiCre-mediated gene disruption, synchronised rings of both integrant SUB2HA3:loxP clones were pulse-treated with rapamycin (RAP) which induces Cre recombinase activity. The parasites were examined ~44 hr later at the multinucleated schizont stage (when SUB2 expression is maximal) within the erythrocytic cycle of RAP treatment (cycle 0). Diagnostic PCR showed efficient excision of the floxed *sub2* sequence (*Figure 1C*), whilst western blot confirmed virtually complete loss of the HA3 signal (*Figure 1D*) and no signal was detectable by IFA in most of the RAP-treated schizonts (*Figure 1E*). No defect in merozoite biogenesis was evident (*Figure 1—figure supplement 1*). These results confirmed disruption of SUB2 expression within a single erythrocytic cycle, and suggested that loss of SUB2 during intracellular development had no impact on schizont maturation.

### SUB2 is the merozoite surface sheddase; loss of SUB2 has no effect on egress but reduces merozoite surface protein shedding and RBC invasion

SUB2 is discharged onto the free merozoite surface to cleave its substrates (*Riglar et al., 2011*; *Harris et al., 2005*; *Howell et al., 2003*; *Green et al., 2006*). We expected egress to be unaffected by loss of SUB2, and in accord with this time-lapse video microscopy showed no differences in rupture of mock- and RAP-treated SUB2HA3:loxP cycle 0 schizonts (*Figure 2A*). To assess the invasive capacity of the released merozoites, highly synchronised schizonts were incubated for 4 hr with RBCs to allow egress and invasion. Newly-invaded (cycle 1) rings were produced in the ΔSUB2 cultures, but at levels only ~50% of those in control cultures (*Figure 2B*). Western blot analysis with selective antibodies of medium harvested from the ΔSUB2 cultures showed substantially decreased levels of MSP1, AMA1, and PTRAMP, consistent with loss of shedding (*Figure 2C,D*). SUB2-mediated shedding during invasion is highly efficient, so antibodies specific to shed segments of MSP1 and AMA1 are invariably unreactive with newly-invaded rings (*Blackman et al., 1996*; *Blackman et al., 1990*; *Blackman et al., 1991*; *Howell et al., 2005*). In contrast, the membrane-proximal AMA1 'stub' and GPI-anchored MSP1$_{19}$ domain that remain on the parasite surface following cleavage at the respective juxtamembrane sites are readily detected in rings (*Blackman et al., 1990*; *Blackman et al., 1991*; *Howell et al., 2003*; *Olivieri et al., 2011*). We therefore used selected antibodies to examine the cycle one rings by IFA. Monoclonal antibody (mAb) X509, which recognises a shed fragment of the MSP1 complex, did not react with control rings, as expected. In contrast, the mAb strongly recognised the ΔSUB2 cycle one rings (*Figure 2E*), indicating that in the absence of SUB2, unshed MSP1 complex was carried into RBCs on invading merozoites. Similarly, antibodies to the AMA1 ectodomain showed stronger reactivity with ΔSUB2 rings than with control rings (*Figure 2F*).

To interrogate the global effects of SUB2 disruption, culture media harvested following rupture of control and ΔSUB2 schizonts in the presence of fresh RBCs were analysed by quantitative mass spectrometry. This revealed reduced levels of several merozoite surface proteins in the ΔSUB2 egress supernatants, consistent with their reduced shedding (*Figure 3* and *Figure 3—source data*

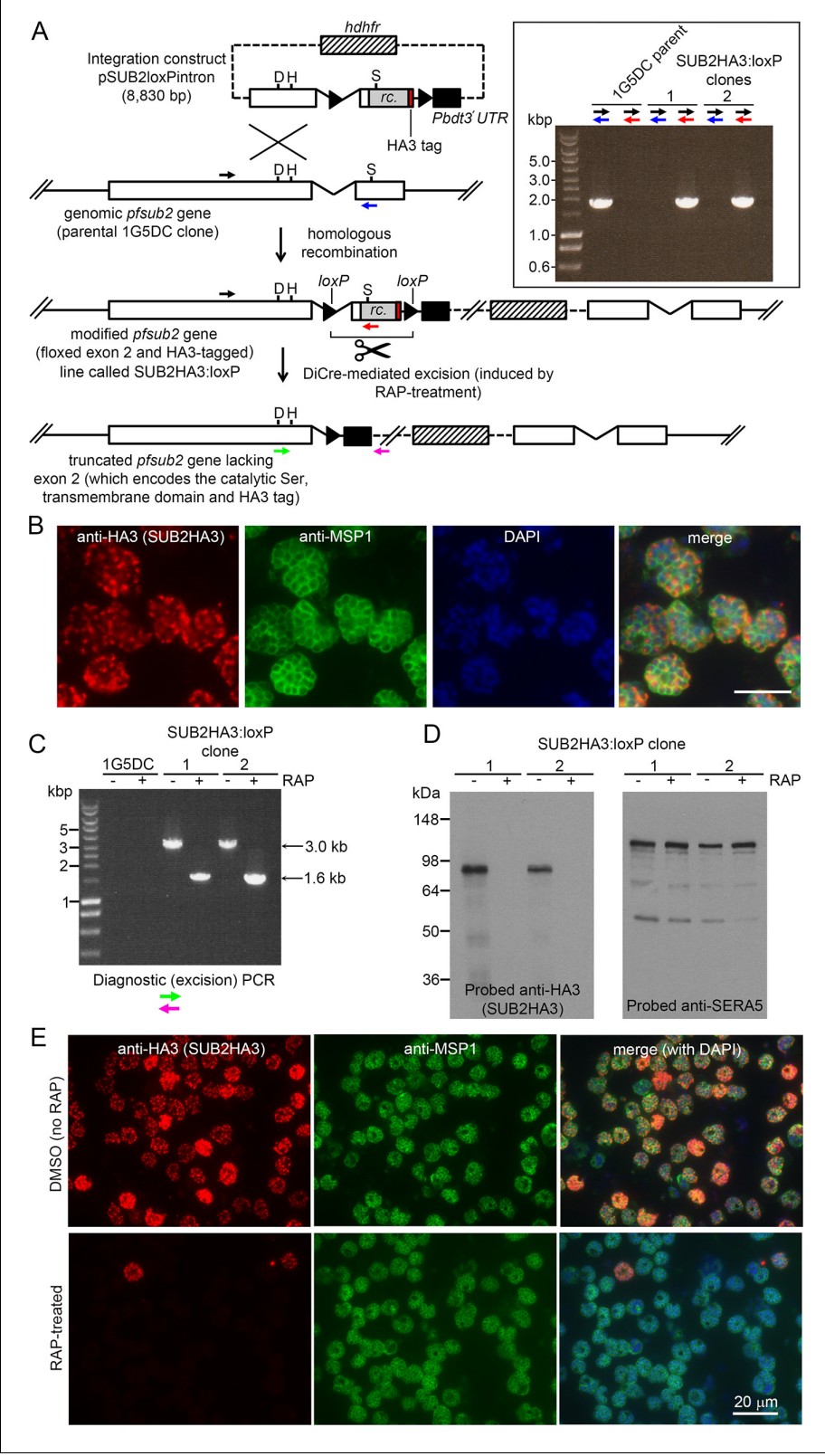

**Figure 1.** Conditional disruption of the *sub2* gene within a single erythrocytic cycle. (**A**) Floxing and epitope tagging. The *P. falciparum sub2* gene (PlasmoDB PF3D7_1136900) comprises two exons and a 143 bp intron. Targeting construct pSUB2loxPintron contains an internal ~1,750 bp gene segment with the intron modified by insertion of a *loxP* site (arrow-head). This was fused to synthetic *sub2*synth sequence (**Harris et al., 2005**) (rc.)
*Figure 1 continued on next page*

*Figure 1 continued*

encoding the TMD and cytoplasmic domain, plus an HA3 tag and stop codon, followed by a second *loxP* site then the 3' UTR of the *P. berghei* dihydrofolate reductase (*dhfr*) gene (Pbdt3' UTR, black box) to ensure correct gene transcription and polyadenylation. The human *dhfr* cassette confers resistance to the antifolate WR99210. Plasmid backbone, dotted line. Single-crossover recombination introduces the entire construct into the genome, floxing exon two and producing a promoterless downstream partial gene duplication. Catalytic triad residues; D, H, S. Coloured arrows; primers for diagnostic PCR (see Materials and methods for sequences and colour codes). Inset, diagnostic PCR confirming pSUB2loxPintron integration. Predicted sizes of PCR amplicons: 1,943 bp (black/blue primers); and 1,954 bp (black/red). (B) IFA localising SUB2HA3 in SUB2HA3:loxP schizonts. Parasite nuclei were stained with 4,6-diamidino-2-phenylindole (DAPI, blue). Scale bar, 10 μm. (C) PCR confirming RAP-induced excision of the floxed exon 2. (D) Western blot of schizont extracts showing loss of SUB2HA3 following RAP treatment. The PV protein SERA5 acted as loading control. (E) IFA showing loss of SUB2HA3 in cycle 0 SUB2HA3: loxP schizonts following RAP treatment of rings. Expression was detectable in just 2 ± 0.1% of RAP-treated schizonts (the microscopic field was deliberately chosen to include 2 of the rare HA3-positive schizonts). See also *Figure 1—figure supplement 1*.

The online version of this article includes the following figure supplement(s) for figure 1:

**Figure supplement 1.** Loss of SUB2 expression in cycle 0 has no impact on schizont development or morphology.

*1*). The depleted proteins included SUB2 itself, as well as AMA1, PTRAMP and members of the MSP1 complex (MSP1, 3, 6 and 7), in accord with the western blot and IFA data. Unexpectedly, the most depleted proteins also included the GPI-anchored merozoite surface proteins Pf92, MSP2, MSP4 and MSP5 (*Gilson et al., 2006*) as well as the MSP7-like protein MSRP2, none of which are components of the MSP1 complex (*Ranjan et al., 2011*; *Lin et al., 2016*; *Stafford et al., 1994*; *Trucco et al., 2001*; *Pachebat et al., 2001*). These results suggest that SUB2 is required to shed these proteins too, indicating a broader repertoire of merozoite surface substrates than previously appreciated.

Collectively, these results confirmed SUB2 as the enzyme responsible for shedding of AMA1, PTRAMP and the MSP1 complex as well as several other merozoite surface proteins, and revealed an important role for SUB2 in invasion. The findings also proved that complete shedding of merozoite surface MSP1 and AMA1 is not a prerequisite for invasion as the ΔSUB2 parasites formed rings, albeit with reduced efficiency.

## Confirmation of SUB2 essentiality by genetic complementation; ΔSUB2 parasites that can invade undergo developmental arrest

To explore the long-term consequences of SUB2 loss, ring-stage SUB2HA3:loxP parasites were mock- or RAP-treated, dispensed at low density into flat-bottomed microwell plates, and cultured undisturbed. Quantitation of the plaques appearing in the wells (resulting from localised zones of RBC destruction; *Thomas et al., 2016*) revealed that RAP-treated parasites produced significantly fewer plaques than controls (*Figure 4A,B*). Genotyping of parasites expanded from 2 of the few plaques in the RAP-treated cultures showed that they were derived from the small fraction of parasites that failed to undergo excision upon RAP treatment (*Figure 4B* and *Figure 1E*). This suggested that correctly excised parasites lacking an intact *sub2* locus failed to replicate.

To test this, and to establish whether the growth defect in the RAP-treated population was solely due to loss of the *sub2* gene, SUB2HA3:loxP parasites were transfected with plasmid pSUB2-BSD designed for ectopic expression of a full-length synthetic SUB2 gene called $sub2_{synth}$ (*Child et al., 2013*). The plasmid included a blasticidin (BSD) resistance marker and an mCherry expression cassette (*Figure 4C*). Parallel SUB2HA3:loxP cultures were transfected instead with a control plasmid (pDC2-mCherry-MCS) lacking the $sub2_{synth}$ expression cassette. Following BSD selection, both lines were RAP-treated to disrupt the genomic *sub2* locus, then examined by plaque assay. This showed that carriage of the $sub2_{synth}$ transgene, but not the control plasmid, compensated for loss of the genomic *sub2* gene (*Figure 4D*). Parasites from single plaques were then expanded in BSD-containing medium and the resulting clones (all mCherry positive) inspected by diagnostic PCR for excision of the floxed chromosomal *sub2* sequence. Whilst none of 25 viable parasite clones harbouring the control plasmid had undergone excision, 2 of 5 selected clones transfected with pSUB2-BSD had undergone excision and so lacked a functional chromosomal *sub2* gene (*Figure 4E*). Episomal

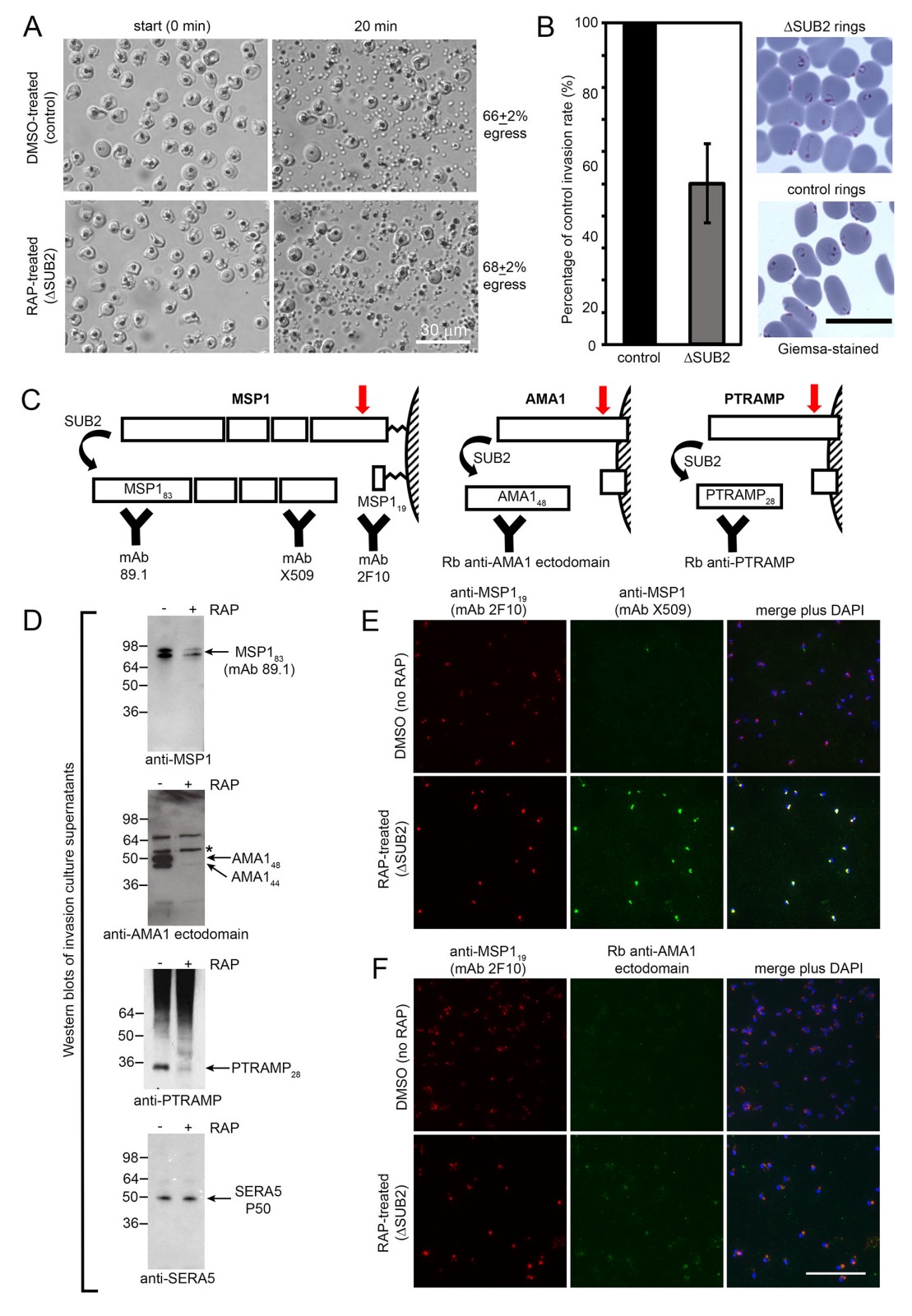

**Figure 2.** SUB2 is not required for egress, but is implicated in invasion and is required for shedding of MSP1, AMA1 and PTRAMP. (**A**) DIC microscopic images showing rupture of control and ΔSUB2 SUB2HA3:loxP schizonts. (**B**) Invasion assay showing ~50% reduction of invasion in ΔSUB2 parasites. Data are presented as percentage of ring formation by control parasites (mean ± SD of 3 biological replicate experiments). Ring parasitaemia in control cultures ranged from 12 to 46%. Right, Giemsa-stained cycle one rings, 4 hr following invasion. (**C**) Schematic showing SUB2-mediated processing of

*Figure 2 continued on next page*

*Figure 2 continued*

the merozoite surface-bound MSP1 complex (previously cleaved into four associated fragments by SUB1), AMA1 and PTRAMP. Red arrow, SUB2 cleavage site. Binding specificity of antibodies used for western blot and IFA are indicated. (**D**) Western blot of culture supernatants following incubation of control and ΔSUB2 SUB2HA3:loxP schizonts with fresh RBCs for 4 hr. Shed proteolytic fragments are indicated. The minor 52 kDa AMA1 species arising from shedding by a merozoite rhomboid-like activity (*Howell et al., 2005*) is asterisked; levels of this cleavage product are not reduced upon SUB2 disruption, as expected. Release of SERA5, which is not processed by SUB2, acts as a loading control. (**E** and **F**) IFA of cycle one rings produced following invasion by control and ΔSUB2 parasites. The MSP1 (**E**) and AMA1 (**F**) ectodomain were detectable only in the ΔSUB2 rings, whilst all rings were reactive with the anti-MSP1$_{19}$ mAb 2F10. Scale bars, 20 μm.

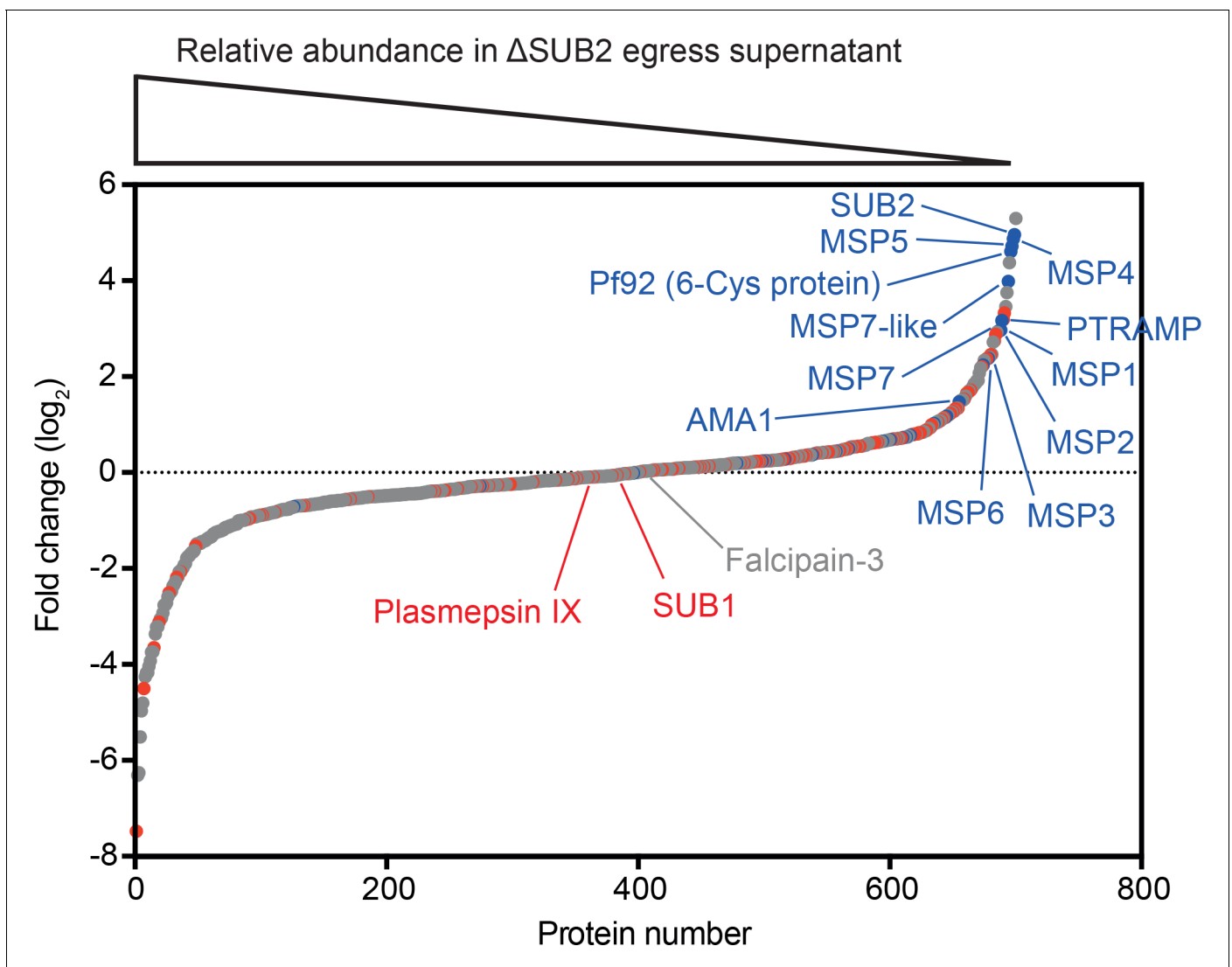

**Figure 3.** Quantitative proteomics shows that SUB2 sheds multiple merozoite surface proteins. S-curve showing comparative mass spectrometric quantitation of *P. falciparum* proteins in egress and invasion supernatants of ΔSUB2 and control SUB2HA3:loxP schizonts. A total of 700 parasite proteins were detected with high confidence. Proteins with predicted secretory signal peptides are indicated in red, established merozoite surface proteins are indicated in blue (these also have signal peptides), whilst other proteins are in grey. Of the top 24 proteins that differed most in abundance between the samples, 12 are known merozoite surface proteins (indicated). SUB1, the intracellular cysteine protease falcipain-3 and the rhoptry-localised aspartic protease plasmepsin IX are highlighted as examples of proteins that did not differ in abundance between samples, as expected. See *Figure 3—source data 1* for mass spectrometric data.

The online version of this article includes the following source data for figure 3:

**Source data 1.** Quantitative mass spectrometric analysis of *P. falciparum* proteins in egress supernatants of ΔSUB2 SUBHA3:loxP schizonts compared with control schizonts.

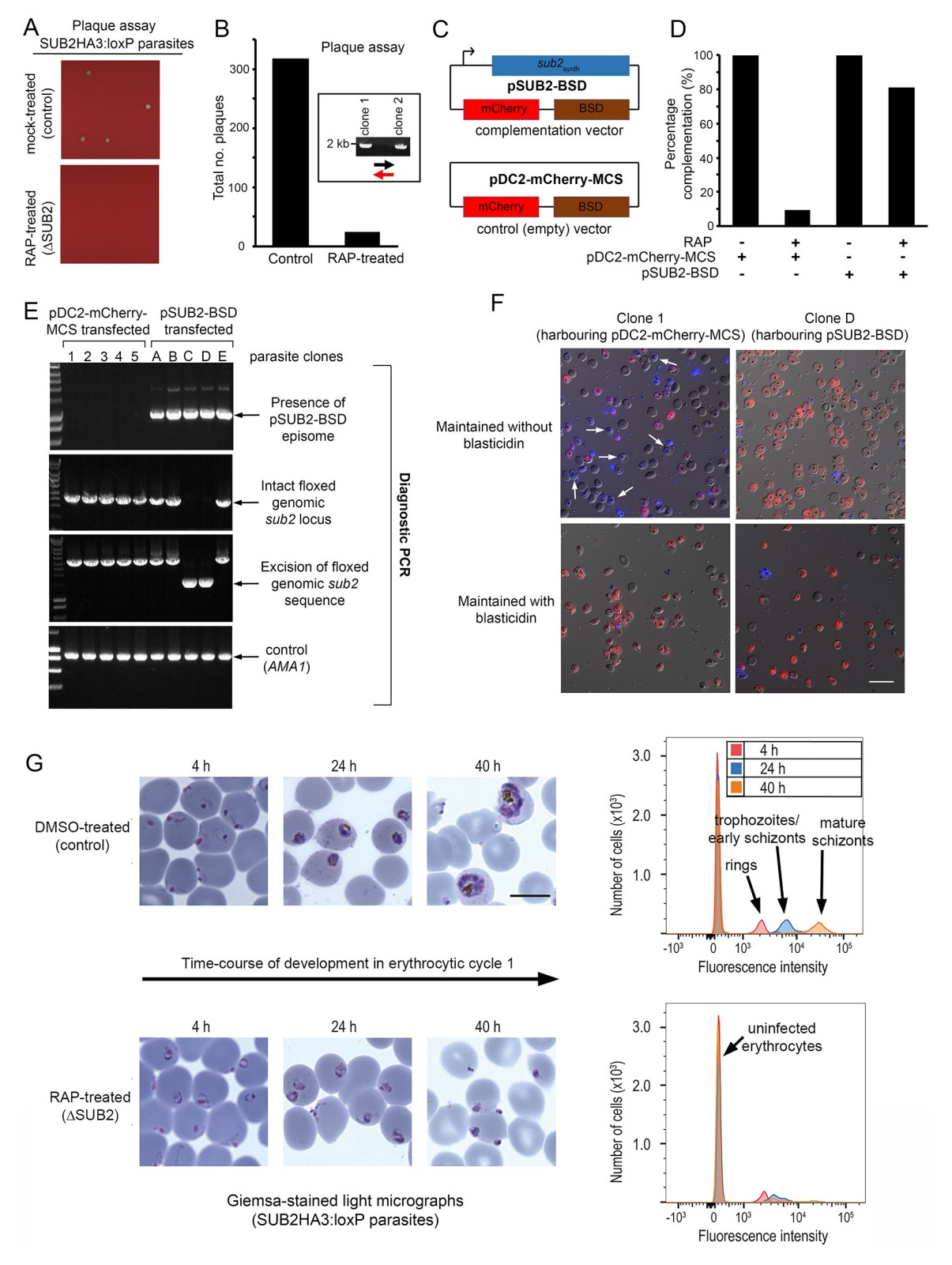

**Figure 4.** ΔSUB2 parasites display a developmental defect in the erythrocytic cycle following gene disruption that can be rescued by genetic complementation. (**A**) Magnified views of plaque assay wells imaged 14 days after plating equal parasite densities (~10 parasites per well). (**B**) Plaque numbers in a total of 60 wells per treatment, showing effects of SUB2 disruption. Two plaques from RAP-treated wells were expanded and the parasites confirmed by PCR to be non-excised. See Materials and methods for primer colour codes. (**C**) Complementation plasmids. Expression of the $sub2_{synth}$

*Figure 4 continued on next page*

Figure 4 continued

gene in pSUB2-BSD was driven by a 5' flanking region that correctly regulates SUB2 expression (*Child et al., 2013*). (D) Genetic complementation of the ΔSUB2 growth defect. Complementation was calculated as the number of plaques obtained in the RAP-treated cultures expressed as a percentage of those in control cultures (100%). (E) Genotyping of parasite clones expanded from the plaque assay in (D). Amplicons diagnostic of indicated genotypes, arrowed. Clones 1–5 (transfected with control plasmid pDC2-mCherry-MCS) and clones A, B and E all possessed an intact genomic *sub2* locus. Clones C and D lacked an intact chromosomal *sub2* gene due to excision, but harboured pSUB2-BSD. (F) Dual DIC/fluorescence images of live Hoechst 33342-stained (blue) parasites of clones one and D (from panel E) following 4 weeks of maintenance. Without BSD, clone one began to lose pDC2-mCherry-MCS as indicated by loss of mCherry (red: examples arrowed), whilst pSUB2-BSD was maintained in all clone D parasites. Scale bar, 20 μm. (G) Parasite development in cycle one following mock or RAP treatment. Arrest of ΔSUB2 parasites was evident within 24 hr. Scale bar, 10 μm. Right: flow cytometry confirms a developmental defect in ΔSUB2 parasites. Parasites were stained with SYBR Green I at indicated times post-invasion and $10^5$ cells analysed. See also *Figure 4—figure supplement 1*, *Figure 4—video 1* and *Figure 4—video 2*.

The online version of this article includes the following video and figure supplement(s) for figure 4:

**Figure supplement 1.** Light and transmission electron microscopic (TEM) analysis of newly-invaded intraerythrocytic ΔSUB2 cycle one parasites shows no discernible structural defect.

**Figure 4—video 1.** Serial block-face scanning electron microscopy (SBF-SEM) of newly-invaded rings (control).

https://elifesciences.org/articles/61121#fig4video1

**Figure 4—video 2.** Serial block-face scanning electron microscopy (SBF-SEM) of newly-invaded rings (ΔSUB2).

https://elifesciences.org/articles/61121#fig4video2

plasmids segregate inefficiently in *Plasmodium* (*O'Donnell et al., 2001*; *O'Donnell et al., 2002*; *van Dijk et al., 1997*) so are usually quickly lost upon removal of drug selection. However, upon further culture of the excised, pSUB2-BSD-harbouring parasites in the absence of BSD, mCherry expression was uniformly maintained (*Figure 4F*, top right) indicating that survival was dependent upon maintenance of the pSUB2-BSD expression plasmid. In contrast, the non-excised clones harbouring the control plasmid began to lose mCherry expression in the absence of BSD (*Figure 4F*, top left image). These results showed that parasite viability requires a functional *sub2* gene.

To understand the loss of viability in ΔSUB2 parasites, we returned to examine the fate of those ΔSUB2 parasites that successfully invaded at the end of cycle 0. Microscopy and flow cytometry revealed an arrest in intracellular development of the cycle one parasites, which - despite appearing initially normal - failed to mature and showed no signs of haemozoin production (a byproduct of haemoglobin digestion) (*Figure 4G* and *Figure 4—figure supplement 1*). Examination by transmission and serial block-face scanning EM identified no structural defects in the ΔSUB2 cycle one rings (*Figure 4—figure supplement 1*, *Figure 4—video 1* and *Figure 4—video 2*). We concluded that the lethal phenotype associated with loss of SUB2 arose from a reduced capacity to invade and a developmental arrest in those parasites that did invade.

## ΔSUB2 merozoites lyse target RBCs

During invasion assays involving ΔSUB2 parasites, we noticed that the culture media were unusually red in colour, suggesting a high free haemoglobin (Hb) content. To explore this, mature cycle 0 schizonts of DMSO- or RAP-treated SUB2HA3:loxP parasites were incubated with fresh RBCs for 4 hr to allow egress and invasion. Levels of Hb in culture supernatants were then quantified by spectrophotometry and SDS-PAGE, and the cells examined by microscopy and flow cytometry (*Figure 5A*). As previously noted, ring production from the ΔSUB2 schizonts was reduced compared to controls. Despite this, higher levels of extracellular Hb appeared in the ΔSUB2 culture supernatants (*Figure 5B*). This did not derive from schizont rupture per se, as incubation of similar numbers of schizonts without addition of RBCs resulted in low levels of Hb release that did not differ between ΔSUB2 and control schizonts. Moreover, Hb release required extensive interaction between released merozoites and the host cells, since it did not occur in the presence of cytochalasin D (cytD), an actin-binding drug that blocks invasion downstream of TJ formation by disrupting the parasite actinomyosin motor that drives invasion (*Miller et al., 1979*; *Figure 5C* and *Figure 5—figure supplement 1*). We conjectured that those ΔSUB2 merozoites unable to invade instead interacted with target RBCs in an abortive manner that led to lysis. To investigate this, following co-incubation of schizonts with RBCs, cultures were supplemented with fluorescent phalloidin, a membrane-impermeable peptide that binds the F-actin of the RBC cytoskeleton (*Atkinson et al., 1982*; *Glushakova et al., 2010*), then immediately examined by live fluorescence microscopy. This revealed

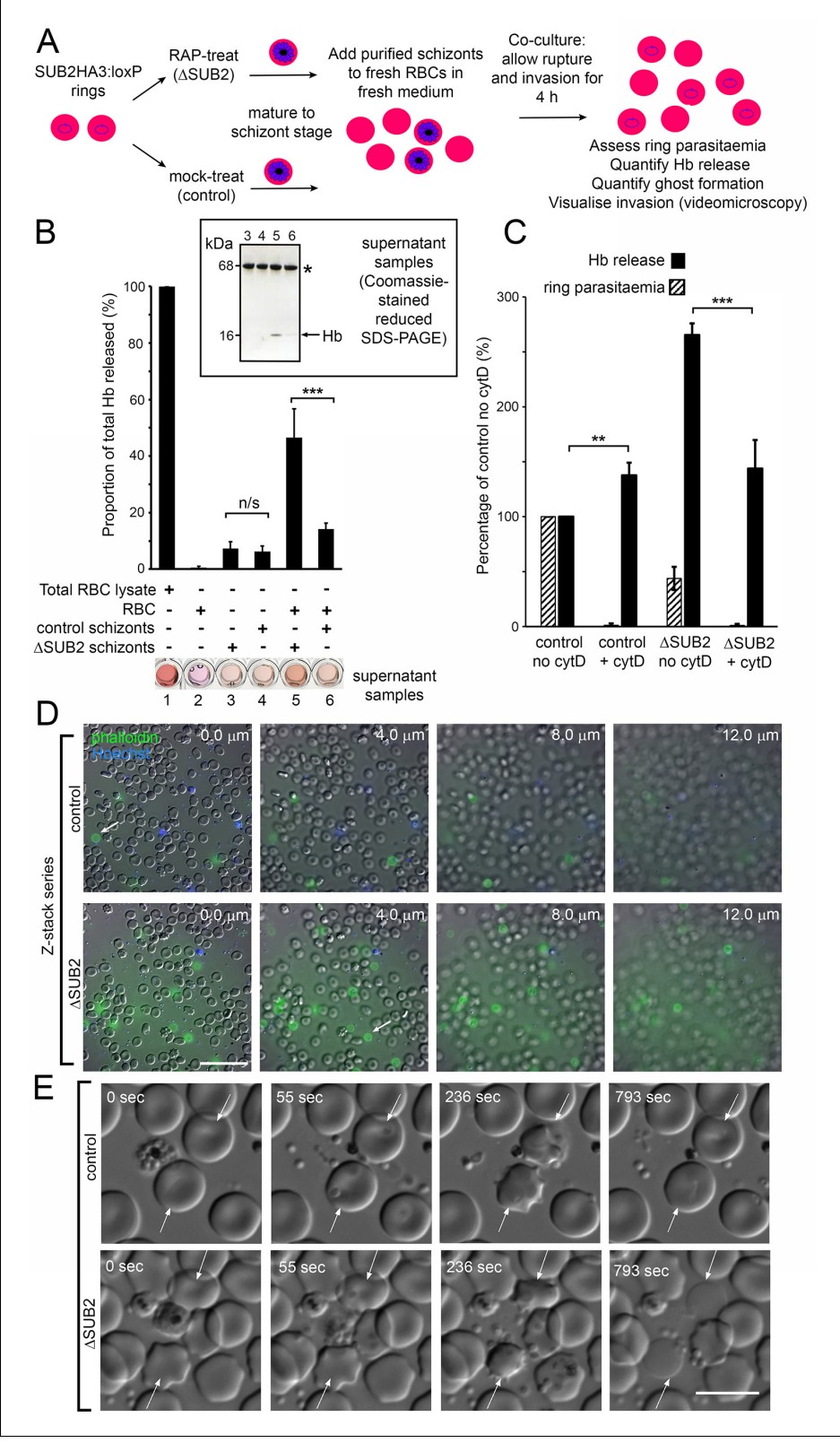

**Figure 5.** Interaction of ΔSUB2 merozoites with target RBCs induces rapid RBC lysis. (**A**) Experimental strategy to quantify the relationship between invasion and Hb release. (**B**) Quantitation of Hb in culture supernatants following egress and invasion. Values shown (averages from three independent experiments) were determined by absorbance at 405 nm, normalised to hypotonic lysates of an equivalent number of uninfected RBCs (100%; see

*Figure 5 continued on next page*

*Figure 5 continued*

Materials and methods). These experiments used high starting schizont parasitaemia (10–15%) to enhance detection of released Hb. Mean ring parasitaemia values following 4 hr of ΔSUB2 schizont/RBC co-culture were only 65.0 ± 4.8% of those in the control schizont/RBC co-cultures, whereas released Hb levels in the ΔSUB2/RBC co-cultures were 3.3 ± 0.7 fold higher than in the control schizont/RBC co-cultures (***). Error bars,± SD. Significance determined by two-tailed unpaired t-test: p=0.006, ***; n/s, p=0.560. Below, microplate wells containing culture supernatants (200 µl) compared to the RBC lysate, showing the high free Hb levels in the ΔSUB2 schizont/RBC co-cultures. Inset: SDS-PAGE analysis of supernatants 3–6. (C) Hb release is prevented by cytD. Ring parasitaemia and released Hb levels relative to those obtained in co-cultures of control schizonts in the absence of cytD (defined as 100%). Whilst Hb release is enhanced throughout by cytD (see also *Figure 5—figure supplement 1*), the drug blocks the high levels of Hb release observed in the ΔSUB2 co-cultures. Shown are averages from two independent experiments. Error bars,± SD. Significance levels determined by two-tailed unpaired t-test: p=0.03, ***; p=0.04, **. (D) Interaction of ΔSUB2 merozoites with RBCs leads to abundant ghost formation. Cultures as in (A) were supplemented with Hoechst 33342 (blue) and Alexa Fluor 488 phalloidin (green) then imaged by dual DIC/fluorescence microscopy. Images are selected from a Z-stack series of single fields, taken at 2.0 µm intervals to detect phalloidin-labelled RBC ghosts, most of which 'float above' the focal plane of the majority of intact RBCs. Ghosts were ~6 fold more numerous in the ΔSUB2 cultures (15.7% of the total RBC count in ΔSUB2 cultures; 2.6% in control cultures). Scale bar, 50 µm. (E) Abortive invasion by ΔSUB2 merozoites leads to RBC lysis. Images from live time-lapse DIC microscopy of egress and subsequent events. Time following start of imaging is indicated. Interaction with target RBCs (arrowed) results in transient echinocytosis, followed by lysis in the case of the ΔSUB2 parasites. Scale bar, 10 µm. See also *Figure 5—video 1*.

The online version of this article includes the following video and figure supplement(s) for figure 5:

**Figure supplement 1.** Cytochalasin D induces low-level haemoglobin release from uninfected human RBCs.
**Figure 5—video 1.** Abortive invasion by ΔSUB2 merozoites leads to rapid target RBC lysis.
https://elifesciences.org/articles/61121#fig5video1

---

rounded, phalloidin-labelled cells that were ~6 fold more abundant in the ΔSUB2 cultures (*Figure 5D*). The dimensions of these cells, their low buoyant density (shown by a tendency to 'float' above intact RBCs) and their accessibility to phalloidin, suggested that they were 'ghosts' derived from lysis of RBCs. We concluded that these derived from abortive invasion attempts in which interaction with ΔSUB2 merozoites had induced RBC lysis.

To directly visualise the fate of target RBCs, we examined the process by live microscopy (*Figure 5E* and *Figure 5—video 1*). Initial interactions with ΔSUB2 merozoites appeared normal, with RBC deformation often followed by parasite entry and echinocytosis. Subsequently however, and in contrast to the behaviour typical of RBCs invaded by control merozoites where echinocytosis quickly resolved, RBCs invaded by ΔSUB2 merozoites often remained rounded and then lysed, as indicated by loss of differential interference contrast (DIC). Lysis often occurred within ~14 min of the initial interaction and was sometimes accompanied by ejection of the merozoite. In other cases lysis took longer and was visualised by images captured over 30–60 min. However, washing and re-culture of the cycle one rings forms resulted in no further Hb release, indicating that lysis was usually rapid. Attempts to fix ΔSUB2 merozoites in the act of invasion for analysis by EM were unsuccessful, possibly due to its transient nature. Nonetheless we concluded that loss of SUB2 led in ~50% of invasion attempts to abortive invasion that culminated in rapid RBC lysis, presumably due to a defect in RBC sealing.

## Inhibition of MSP1 shedding by cleavage site mutagenesis inhibits intracellular parasite development

To dissect the ΔSUB2 phenotype, we investigated the effects of mutations that directly prevent SUB2-mediated shedding of the most abundant SUB2 substrate, the MSP1 complex. We previously mapped the SUB2 cleavage site in MSP1 to the Leu1606-Asn1607 bond just upstream of the C-terminal MSP1$_{19}$ domain (*Blackman et al., 1990*) (see also *Figure 2C*) and showed that proline substitutions of the residues flanking the AMA1 SUB2 cleavage site blocks cleavage (*Olivieri et al., 2011*). Based on this, transgenic parasite line iMSP1$_{PP}$ was generated in which DiCre-mediated excision of a floxed segment of the *msp1* gene produced a partial allelic replacement, substituting the Leu-Asn cleavage motif with a Pro-Pro motif (*Figure 6A* and *Figure 6—figure supplement 1*). In a control parasite line, iMSP1$_{LN}$, excision reconstituted the wild type motif.

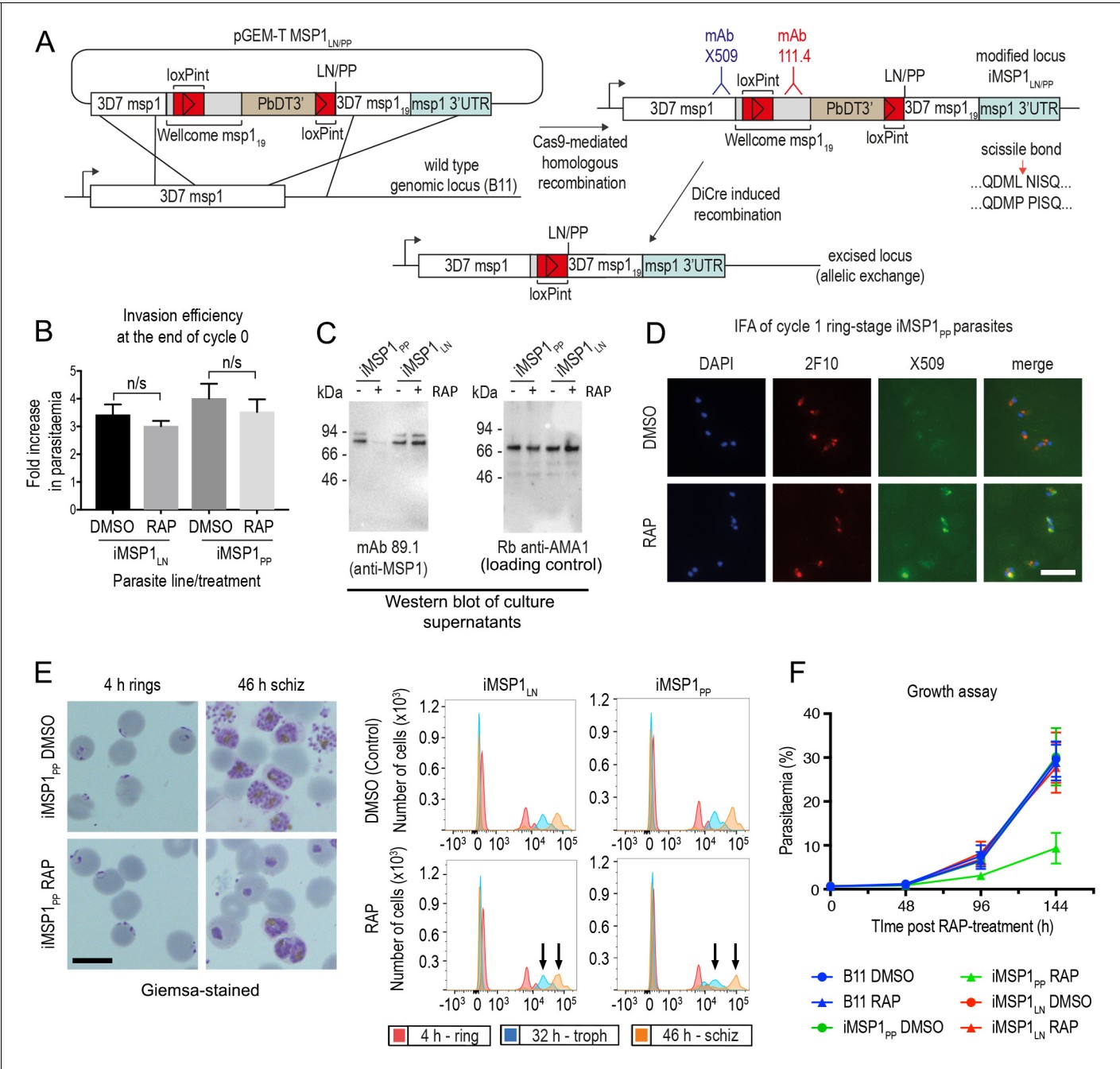

**Figure 6.** Inhibition of MSP1 shedding by conditional mutagenesis of the SUB2 cleavage site inhibits cycle one intracellular parasite development. (A) Strategy used for conditional mutagenesis of the SUB2 processing site in MSP1 (see also *Figure 6—figure supplement 1*). (B) Ring formation is unaffected by conditional mutagenesis of the MSP1 SUB2 processing site. Values averaged from three independent experiments. Error bars,± SD. Significance determined by two-tailed t-test: p values for the comparison of DMSO and RAP-treated MSP1$_{LN}$ and MSP1$_{PP}$ lines were 0.17 and 0.30, respectively (n/s). (C) Reduced MSP1 shedding during invasion by RAP-treated iMSP1$_{PP}$ parasites. (D) IFA showing that cycle one rings formed by RAP-treated iMSP1$_{PP}$ parasites were recognized by mAb X509, consistent with reduced MSP1 shedding. All rings are detected by the MSP1$_{19}$-specific mAb 2F10. (E) Reduced MSP1 shedding affects intracellular development. Control or RAP-treated iMSP1$_{LN}$ and iMSP1$_{PP}$ schizonts were incubated with RBCs and the cycle one rings monitored. Microscopy shows retarded ring development in RAP-treated iMSP1$_{PP}$ parasites, corroborated by flow cytometry (measuring parasite DNA content) showing 23.6 ± 5.0% reduced replication by 46 hr compared to similarly treated iMSP1$_{LN}$ controls (compare arrowed peaks). Scale bars, 10 μm. (F) Reduced replication of RAP-treated iMSP1$_{PP}$ parasites.

The online version of this article includes the following figure supplement(s) for figure 6:

**Figure supplement 1.** Conditional mutagenesis of the *P. falciparum msp1* gene to selectively block SUB2-mediated shedding of MSP1.

RAP-treatment of iMSP1$_{PP}$ and iMSP1$_{LN}$ parasite clones produced the expected genomic excision events (*Figure 6—figure supplement 1*). To examine the effects of the Leu-Asn to Pro-Pro substitution, synchronous rings were RAP- or mock-treated, matured to schizont stage, then allowed to undergo egress in the presence of RBCs. This showed no significant effects on efficiency of ring formation (*Figure 6B*), although examination of culture supernatants following invasion showed a selective reduction in MSP1 shedding in RAP-treated iMSP1$_{PP}$ cultures, consistent with the predicted effects of the mutations on SUB2-mediated cleavage (*Figure 6C*). Cycle one rings from RAP-treated iMSP1$_{PP}$ schizonts were strongly recognised by mAb X509, similar to ΔSUB2 cycle one rings (*Figure 6D*), indicating successful invasion despite reduced MSP1 shedding. The newly-invaded cycle one mutant iMSP1$_{PP}$ rings appeared morphologically normal, but further monitoring revealed extensive retarded intracellular development (*Figure 6E*), and longer-term experiments confirmed a replication defect in the RAP-treated iMSP1$_{PP}$ parasites (*Figure 6F*). We concluded that selective inhibition of MSP1 shedding at invasion did not affect RBC entry but led to defective intracellular parasite development similar to that although not as severe as in the ΔSUB2 mutants.

## Inhibition of AMA1 shedding by cleavage site mutagenesis has no impact on invasion but loss of AMA1 expression prevents invasion and results in host RBC lysis

The MSP1 mutagenesis data provided a plausible explanation for the impact of SUB2 loss on intracellular parasite growth, but did not explain the invasion phenotype of ΔSUB2 merozoites nor their capacity to lyse RBCs. Prior to this work, the only other experimentally-demonstrated essential SUB2 substrate was the microneme protein AMA1. The AMA1 ectodomain comprises three globular domains linked to the TMD via a short juxtamembrane segment (*Pizarro et al., 2005*). Cleavage of *P. falciparum* AMA1 by SUB2 occurs at the Thr517-Ser518 bond 29 residues upstream of the TMD (*Figure 2C*), releasing domains I-III, leaving a juxtamembrane 'stub' bound to the merozoite surface (*Howell et al., 2003*). AMA1 can additionally be shed by the parasite rhomboid protease ROM4 via cleavage within the TMD at the Ala550-Ser551 bond (*Howell et al., 2003*; *Howell et al., 2005*); this can be blocked by a Tyr substitution of Ala550, a mutation tolerated by the parasite (*Olivieri et al., 2011*). To probe the biological significance of SUB2-mediated shedding of AMA1, we conditionally modified both the SUB2 and ROM4 cleavage sites to simultaneously render them refractory to cleavage. For this, we generated transgenic parasite line iΔRΔS in which excision of a floxed *ama1* gene simultaneously substitutes the Thr517-Ser518 site with a Pro-Pro motif and Ala550 with a Tyr residue (*Olivieri et al., 2011*; *Figure 7* and *Figure 7—figure supplement 1*).

Replication of untreated iΔRΔS parasites was indistinguishable from parental parasites, and RAP treatment produced the expected genomic changes (*Figure 7—figure supplement 1*) as well as the anticipated decrease in AMA1 shedding (*Figure 7A*). However, we detected no change in the invasive capacity of the RAP-treated parasites and replication was unaffected (*Figure 7B*). Given the absence of deleterious effects, we reasoned that direct mutagenesis of the AMA1 cleavage sites should generate viable parasites, so we used targeted homologous recombination to directly introduce the same cleavage site mutations into the *ama1* gene (*Figure 7—figure supplement 1*). The resulting parasite line, dΔRΔS, displayed a phenotype similar to that of the RAP-treated iΔRΔS parasites, with reduced AMA1 shedding (*Figure 7A*) but no effects on invasion or replication (*Figure 7B*). These results suggested that inhibition of AMA1 shedding per se was not responsible for the invasion and lysis phenotype observed in the ΔSUB2 mutants.

Given the role of AMA1 in TJ formation, we examined whether the ΔSUB2 phenotype might in part reflect loss of AMA1 function. We generated a parasite line (iΔAMA1) in which DiCre-mediated excision ablated AMA1 expression by severely truncating the gene (*Figure 7—figure supplement 2*). In a control line (iAMA1_C), excision reconstituted a functional full-length gene (*Figure 7—figure supplement 2*). RAP treatment of the parasites produced the expected genomic modifications (*Figure 7—figure supplement 2*), with loss of AMA1 expression in mature cycle 0 schizonts of the iΔAMA1 clones (*Figure 7C,D*). This led to complete loss of ring formation and proliferation following rupture of RAP-treated iΔAMA1 schizonts at the end of cycle 0 (*Figure 7E,F*), confirming AMA1 essentiality. Despite the lack of invasion, levels of Hb released into the culture media were much higher in RAP-treated iΔAMA1 schizonts incubated with RBCs than in controls (*Figure 7G*). To determine the source of the Hb, we used microscopy to visualise interactions between ΔAMA1 merozoites and RBCs. This confirmed the loss of invasion, whilst showing that interaction of ΔAMA1 merozoites

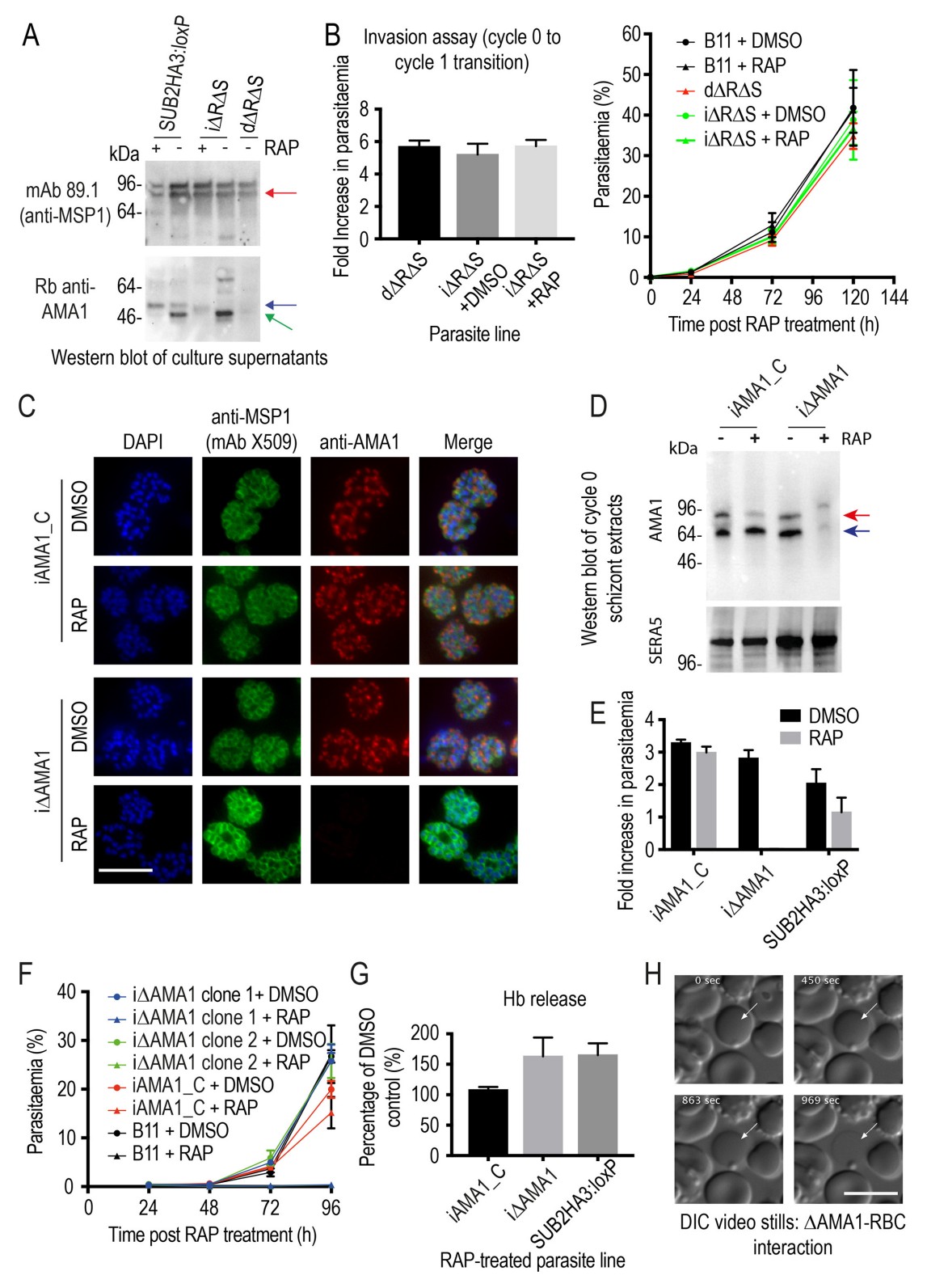

**Figure 7.** Inhibition of AMA1 shedding by cleavage site mutagenesis has no effect on invasion, but AMA1 disruption prevents invasion and leads to RBC lysis. (A) Selective reduction in AMA1 shedding in dΔRΔS and RAP-treated iΔRΔS parasites. Shed fragments are arrowed (MSP1$_{83}$, red; AMA1$_{52}$, blue; AMA1$_{48}$, green). (B) Cleavage site mutagenesis has no impact on invasion or parasite replication. Data in both cases are averages of 3 independent experiments. Error bars,± SD. (C) IFA showing loss of AMA1 expression in RAP-treated iΔAMA1 cycle 0 schizonts. (D) Disruption of AMA1

*Figure 7 continued on next page*

*Figure 7 continued*

expression (arrowed) in RAP-treated iΔAMA1 parasites. Antibodies to SERA5 were used as a loading control. (**E**) Invasion assay showing complete loss of invasion following rupture of RAP-treated cycle 0 iΔAMA1 schizonts in the presence of RBCs. (**F**) Loss of replication of RAP-treated iΔAMA1 parasite clones. (**G**) Increased Hb release upon interaction of ΔAMA1 parasites with RBCs is similar to that in RAP-treated SUB2HA3:loxP (ΔSUB2) parasites. (**H**) Time-lapse DIC microscopy images of interactions between a ΔAMA1 merozoite and an RBC (arrowed), showing eventual lysis of the RBC. Time following start of imaging is indicated. Scale bars, 10 µm. See also *Figure 7—figure supplement 1*, *Figure 7—figure supplement 2* and *Figure 7— video 1*.

The online version of this article includes the following video and figure supplement(s) for figure 7:

**Figure supplement 1.** Conditional or direct mutagenesis of the *P. falciparum ama1* gene to selectively block proteolytic shedding of AMA1.
**Figure supplement 2.** Conditional disruption of the *P. falciparum ama1* gene.
**Figure 7—video 1.** Interactions between ΔAMA1 merozoites and target RBCs lead to RBC lysis.
https://elifesciences.org/articles/61121#fig7video1

with target RBCs produced extensive RBC echinocytosis (*Figure 7—video 1*). We did not observe rapid RBC lysis; however, target RBCs often transformed to a rounded form, remaining in this state for some time before losing Hb content (*Figure 7H*). We concluded that loss of AMA1 not only prevented invasion but also produced host RBC lysis upon merozoite interaction, presumably due to an RBC sealing defect.

## Discussion

We have provided the first genetic proof of SUB2 as the merozoite sheddase, and have shown that abolishing SUB2-dependent cleavage produces a complex phenotype that is lethal in at least two superficially distinct manners; merozoites that successfully complete invasion fail to develop, whilst other merozoites induce RBC lysis at or shortly following entry. Our most important conclusion is that SUB2-mediated protein shedding is essential for parasite survival. Our primary mechanistic explanation for this is that shedding is required for resealing of the RBC membrane at invasion (*Figure 8*).

Using IFA, western blot and mass spectrometry to compare SUB2-expressing and ΔSUB2 parasite cultures, we confirmed that SUB2 is required for shedding of MSP1, AMA1 and PTRAMP. We also identified several putative new SUB2 substrates not known to associate with any of the three previously known substrates. Some of the most prominent of these, including MSP2, MSP4, MSP5 and Pf92, belong to a group of merozoite surface proteins known or predicted to possess GPI anchors (*Gilson et al., 2006*; *Sanders et al., 2005*; *Marshall et al., 1998*; *Marshall et al., 1997*). Since GPI-anchored proteins cannot be shed by rhomboid cleavage (which occurs within TMDs), it is likely that shedding of each is independently catalysed by SUB2. The other newly identified SUB2 substrate, MSRP2, likely has no GPI anchor (so could be peripherally associated with one of the GPI-anchored substrates) but does undergo proteolytic cleavage around egress (*Kadekoppala et al., 2010*). This wide range of SUB2 substrates was unexpected; indeed earlier work that did not benefit from access to mutants concluded that MSP2 and MSP4 are not shed at invasion (*Boyle et al., 2014*). Previous extensive mutagenesis analysis of the AMA1 SUB2 cleavage site (*Olivieri et al., 2011*) indicated that SUB2 has a rather promiscuous substrate recognition profile, cleaving its membrane-bound substrates at a roughly similar distance from the membrane rather than at a specific amino acid sequence. Our new mass spectrometric evidence revealing a diverse range of additional putative SUB2 substrates supports that model and underlines the potential for multiple consequences of SUB2 inhibition. It was interesting to note that SUB2 depletion resulted in significantly *upregulated* levels in culture supernatants of a number of merozoite proteins, particularly prominent examples including a conserved protein of unknown function (PF3D7_0520800), an inner membrane complex protein (PF3D7_1003600), and the myosin A tail interacting protein PF3D7_1246400 (MTIP), a component of the merozoite actinomyosin contractile system that drives invasion (see *Figure 3—source data 1* and the left-hand side of the plot in *Figure 3*). Whilst we were initially puzzled by this, we suspect that these proteins might simply be derived from degradation of merozoites that failed to

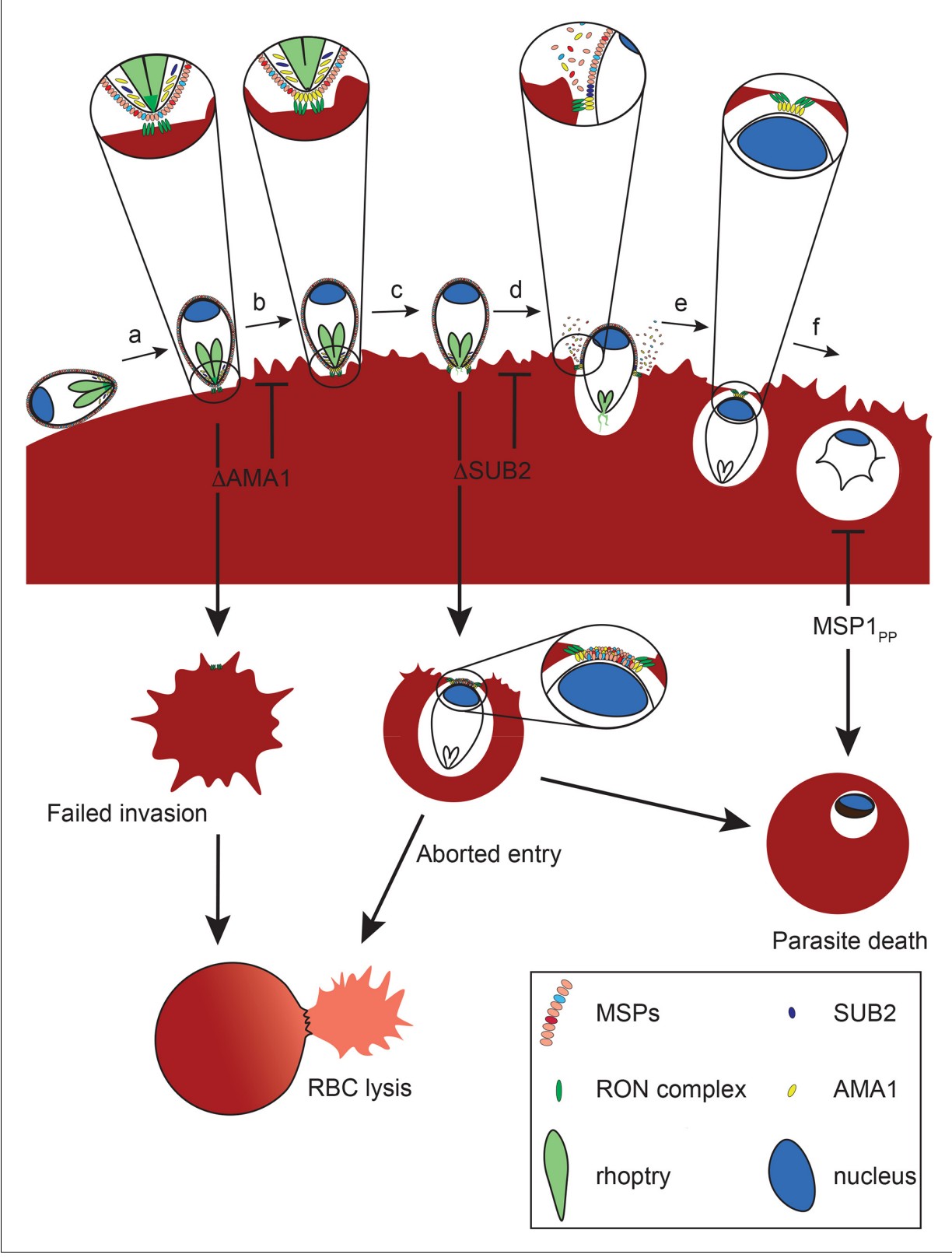

**Figure 8.** Model of merozoite-RBC interactions following disruption of SUB2 or AMA1, or selective ablation of MSP1 shedding (MSP1$_{PP}$). Following the initial merozoite-RBC interaction, reorientation (**a**) brings the merozoite apical prominence into contact with the RBC surface. Rhoptry neck proteins are discharged into the RBC membrane (**b**), inducing echinocytosis. AMA1 is released from micronemes to interact with the RON complex, forming the TJ through which the merozoite begins to enter the nascent PV (**c**). During entry, several protein components of the merozoite surface coat are shed by

*Figure 8 continued on next page*

*Figure 8 continued*

SUB2 (d). As the merozoite is internalised the RBC membranes are drawn together at the posterior end by the TJ (e). The opposed membranes finally seal and undergo scission, fully encapsulating the parasite within the cell (f). In the absence of AMA1 (ΔAMA1), insertion of the RON complex results in echinocytosis but without formation of the TJ. The RBC membrane remains destabilised, leading to RBC lysis. Loss of SUB2 expression (ΔSUB2) leads to a plug of unshed surface proteins that prevent RBC membrane sealing, leading to abortive entry and RBC lysis. Some ΔSUB2 parasites manage to enter fully but subsequently fail to develop beyond ring stage, perhaps due to a defect in DV biogenesis because of an accumulation of multiple unshed proteins (or one or more critical proteins) on the plasma membrane of the internalised parasite. A similar but less severe intracellular growth phenotype results from selective loss of MSP1 shedding in the MSP1$_{PP}$ mutant.

invade in the ΔSUB2 cultures; these merozoites would accumulate extracellularly and eventually disintegrate, likely leading to increased supernatant levels of merozoite proteins unrelated to the activity of SUB2.

Despite the reduction in merozoite surface protein shedding, ΔSUB2 parasites formed new rings at the end of cycle 0, albeit at ~50% reduced efficiency. These uniformly arrested, and growth assays supported by genetic complementation proved that SUB2 is indispensable for parasite survival. To dissect the phenotype, we generated lines in which shedding of the most abundant SUB2 substrate, the MSP1 complex, was selectively inhibited. These mutants invaded with wild type efficiency but displayed a developmental defect similar to – if somewhat less severe than - that of the ΔSUB2 mutants. Given the technical difficulties in determining causality upon intracellular arrest in *Plasmodium*, we can only speculate on the underlying defect(s). The MSP1$_{19}$ species remaining on the merozoite surface following SUB2-mediated shedding has been shown to be a highly stable constituent of the post-invasion membrane rearrangements involved in formation of the acidic digestive vacuole (DV) that is the site of haemoglobin catabolism during intraerythrocytic development (*Dluzewski et al., 2008*). Formation of the DV appears to arise through endocytosis of plasma membrane constituents (including MSP1$_{19}$), forming a number of small endocytic vesicles that eventually fuse to form the single DV. As a result, MSP1$_{19}$ is the earliest known marker for the DV in the developing ring (*Dluzewski et al., 2008*). We speculate that in the absence of SUB2-mediated shedding, the bulky MSP1 complex remaining on the plasma membrane of the internalised parasite, likely together with other unshed SUB2 substrates, interferes with DV biogenesis, inhibiting haemoglobin digestion and stalling growth. The lack of haemozoin in both the cycle 1 ΔSUB2 parasites and many arrested iMSP1$_{PP}$ mutants is consistent with this model. We suggest that the intracellular death phenotype of the ΔSUB2 mutant was more severe than that of the RAP-treated iMSP1$_{PP}$ mutant due to the fact that in the case of the ΔSUB2 mutant none of the multiple SUB2 substrates are shed, leading to a much more pronounced effect on DV biogenesis than that caused by lack of shedding of the MSP1 complex alone.

The most dramatic phenotype associated with loss of SUB2 was extensive lysis of targeted RBCs. What could underlie this? The mechanisms regulating PVM formation, resealing and scission during invasion by apicomplexan parasites are not understood. Numerous EM studies (e.g. *Aikawa et al., 1978*) indicate that, topologically, invasion resembles induced invagination of the RBC membrane, so that the host RBC cytosol is never exposed to the extracellular milieu. Conflicting with this is recent evidence that successful RBC invasion by *P. falciparum* merozoites involves the formation of a transient discontinuity in the host cell membrane, allowing Ca$^{2+}$ flux into the RBC (*Weiss et al., 2015*; *Volz et al., 2016*). Similarly, an elegant time-resolved patch-clamp study in the related apicomplexan *Toxoplasma gondii* detected a spike in host cell membrane conductance upon apical attachment of the parasite, again consistent with pore formation (*Suss-Toby et al., 1996*). Near the end of invasion, as the parasite posterior enters the nascent PVM, it seems likely that a membrane scission event seals the shrinking TJ aperture, and evidence for this too was found in the *Toxoplasma* study (*Suss-Toby et al., 1996*). Recent work in *Toxoplasma* suggests that PVM scission is aided by rotation of the parasite along its apical-posterior axis immediately following entry (*Pavlou et al., 2018*), and malaria merozoites have also been observed to spin post invasion (*Dvorak et al., 1975*; *Yahata et al., 2012*).

Given these models of pore formation early in invasion and a membrane scission event at the end of the process, it seems plausible that a defect in either could cause a localised loss of host cell membrane integrity sufficiently catastrophic to result in lysis. RBC lysis associated with defective or delayed entry has been previously documented even in wild type *Plasmodium* (*Dvorak et al., 1975*; *Yahata et al., 2012*), and indeed we noticed phalloidin-labelled RBC ghosts at low frequency in our control cultures, so the process is clearly sensitive to perturbation. We suggest that RBC lysis associated with loss of SUB2 is at least in part due to an accumulation of unshed merozoite surface proteins at the narrowing TJ aperture, preventing sealing (*Figure 8*). That lysis does not occur in the presence of cytD, which blocks invasion at the point of TJ assembly (*Miller et al., 1979*), is consistent with this. Our MSP1 mutagenesis results show that inhibition of MSP1 shedding is not alone sufficient to cause lysis, but since MSP1 is just one of many SUB2 substrates this mutant cannot recapitulate the entire ΔSUB2 phenotype, where multiple proteins remain unshed from the invading parasite surface. It is similarly possible that, among the many substrates of SUB2, there is one that captures the entire phenotype. Genetic analysis of each of the new substrates identified here would be required to resolve that question.

While we favour the above model, alternative scenarios cannot be ruled out. Disruption of the *Toxoplasma* rhomboid protease TgROM4, which sheds several key parasite adhesins, resulted in 'hyper adhesive' parasites that bound to host cells with enhanced avidity but displayed a defect in entry (*Buguliskis et al., 2010*; *Rugarabamu et al., 2015*). This was likely due to excessive adhesion and the absence of release of points of contact with the host cell membrane, creating a physical obstacle to entry into the nascent PV. A similar phenomenon, leading to high levels of adhesive traction on the RBC membrane, could potentially underlie the RBC lysis induced by the ΔSUB2 mutants. SUB2 depletion could also conceivably result in a defect in rhoptry discharge, although we consider this unlikely since the ΔSUB2 merozoites induced extensive RBC echinocytosis, a phenomenon associated with normal rhoptry function in *Plasmodium* (*Weiss et al., 2015*).

The ΔSUB2 phenotype may be magnified by the fact that the SUB2 substrate AMA1 is a core component of the TJ, bridging the parasite to the RBC through interactions with RON2, a member of a complex of rhoptry neck proteins inserted into the host cell membrane early in the invasion pathway, perhaps at pore formation (*Alexander et al., 2005*; *Srinivasan et al., 2011*; *Lamarque et al., 2011*; *Tyler and Boothroyd, 2011*; *Lamarque et al., 2014*; *Collins et al., 2009*). The AMA1-RON2 interaction may play a role in drawing the TJ orifice closed as the parasite posterior enters the PV. If so, an attractive concept is that SUB2-mediated release of the AMA1 ectodomain may facilitate pinching off of the PVM, and so we initially considered that loss of AMA1 shedding might be primarily responsible for the ΔSUB2 lysis phenotype. We were therefore surprised to find that selective inhibition of AMA1 shedding had no impact on invasion or viability. By contrast, disruption of AMA1 expression *did* produce a lysis phenotype, as well as the expected loss of invasion. We reconcile these disparate observations by speculating that lysis occurs upon loss of AMA1 because the RBC membrane is compromised early in invasion by insertion of the RON complex (which occurs independently of the presence of AMA1; *Lamarque et al., 2014*; *Giovannini et al., 2011*) but subsequent stabilisation of the TJ cannot occur. We suspect that, whilst superficially similar, the mechanism underlying RBC lysis by ΔAMA merozoites is therefore distinct from that induced by the ΔSUB2 mutant (*Figure 8*).

A role for *P. falciparum* AMA1 in RBC sealing has been postulated previously, based on work using a less efficient conditional mutagenesis strategy which led to variable levels of AMA1 depletion (*Yap et al., 2014*). In that study many parasites that invaded failed to develop, leading the authors to suggest that even a partial sealing defect could stall development. We concur, and although we detected no membrane defects in the ΔSUB2 cycle one rings, the developmental arrest we observed in the ΔSUB2 and MSP1 cleavage site mutants might also stem from a sealing defect insufficiently severe to cause lysis. Interestingly, neither the Yap et al. study nor other studies of *Plasmodium* or *Toxoplasma* AMA1-null mutants noted widespread host cell lysis (*Giovannini et al., 2011*; *Mital et al., 2005*; *Bargieri et al., 2013*). We suspect three likely reasons for this. First, our conditional gene disruption system is highly efficient, with close to 100% conversion in a single erythrocytic cycle. Second, in *Toxoplasma* disruption of AMA1 can lead to upregulation of AMA1 paralogues absent in *Plasmodium* (*Lamarque et al., 2014*; *Parker et al., 2016*). Third, RBCs are more sensitive to lysis resulting from membrane wounding than the nucleated cells invaded by

*Toxoplasma*, perhaps due to the absence in RBCs of endomembrane-dependent repair mechanisms (*McNeil et al., 2003*).

In summary, the lethal consequences of SUB2 disruption likely result from: (1) a combination of loss of shedding of multiple surface proteins, leading to defective RBC sealing; and (2) arrested post-invasion development due to inhibition of DV biogenesis and/or incorrect sealing (*Figure 8*). Our work identifies SUB2 as a critical mediator of host RBC integrity and parasite viability. Drug-like inhibitors of SUB2 protease activity have potential as a new class of antimalarial drug that would inhibit invasion and parasite replication.

## Materials and methods

### Culture and synchronisation of *P. falciparum*

Asexual blood-stages of *P. falciparum* were maintained at 5–10% parasitaemia in RPMI 1640 supplemented with 0.5% Albumax II (RPMI 1640-Albumax; ThermoFisher). Parasites were synchronised at 48–96 hr intervals using standard methodology (*Blackman, 1994*). Briefly, mature schizonts were enriched by centrifugation over a 70% isotonic Percoll cushion (GE Healthcare Life Sciences) and then allowed to invade fresh AB+ RBCs (NHSBT) for 1–2 hr. Following invasion, remaining intact schizonts and schizont debris were removed by centrifugation over a 70% isotonic Percoll cushion and the newly-invaded ring stages further treated with 5% (w/v) sorbitol for 7 min at 37°C to lyse any residual schizonts. The final ring cultures were washed and returned to culture or used as required.

### Construction of plasmid constructs to genetically modify *P. falciparum*

PCR amplicons used in plasmid cloning were generated using Fusion high fidelity DNA polymerase (ThermoFisher) or Platinum Taq DNA polymerase, High Fidelity (ThermoFisher) and purified using Qiagen PCR purification or Qiagen Gel extraction kits.

### Construct for generation of conditional SUB2 disruption parasite line SUB2HA3:loxP

A fragment of *sub2* sequence (S2syn) was commercially synthesised (GeneArt), comprising a stretch of native *P. falciparum* 3D7 *sub2* sequence followed by the 3D7 *sub2* intron containing an internal *loxP* site (*loxPS2Int*) and finally a stretch of recodonised *sub2* gene sequence encoding the 3D7 amino acid sequence but using a different codon usage (*Harris et al., 2005*). The 3' end of the recodonised *sub2* gene was amplified from plasmid psub2-sub2wHA3_F1 (*Child et al., 2013*) using primers sgS2_HBA_F and sgS2_X1_R and cloned into plasmid pHH1_sera5_LoxP1 (*Collins et al., 2013a*) which has a *loxP* site downstream of the Xho I site at the 3' end of the *sera5* gene sequence, using Hpa I and Xho I, generating plasmid pHH1_sgS2_3 HA. A *sub2* target region was amplified from 3D7 genomic DNA using primers S2endo_HpaI_F and S2endo_BspEI_R and cloned into pHH1_sgS2_3 HA using Hpa I and Bsp EI resulting in plasmid pHH1_endo/sgS2_3 HA. Plasmid pHH1_sub2_KO was obtained by cloning S2syn into pHH1_endo/sgS2_3 HA using Bsp EI and Age I. The unmodified endogenous and modified (integrated) *sub2* loci were detected by PCR using forward primer SUB2for_Int and reverse primers SUB2rev_wt and SUB2rev_Int (1.95 kb) respectively. The intact and excised *sub2* loci following RAP-induced recombination were detected with forward primers S2assplasfor2 and nonexSUB2_F, respectively, and reverse primer pHH1R1 (giving rise to 1.7 and 2.6 kb fragments, respectively).

### SUB2 complementation plasmid

The *sub2* expression cassette was excised from plasmid pSUB2_sub2wHA_PEX (*Child et al., 2013*) using Not I and Spa I. This sequence was cloned into pDC2-mCherry-MCS (*Thomas et al., 2018*) pre-digested with Not I and Hpa I giving rise to plasmid pSUB2-BSD. Plasmid pDC2-mCherry-MCS was used as the control plasmid for complementation. The presence of episome was detected by PCR using primers sgSUB2_5 and sgSUB2_31 (amplicon size; 1.0 kb). PCR amplification of the *ama1* gene locus with primers A1seq1for and A1seq6rev was used as a loading control (0.9 kb).

## Construct design to generate transgenic parasite lines iMSP1$_{PP}$ and iMSP1$_{LN}$ for conditional expression of mutant MSP1

Three synthetic gene fragments were generated (GeneArt). These were: (1) Fragment A: chimera of the 3D7 *msp1* sequence and Wellcome *msp1$_{19}$* with an internal *loxPint* sequence. This sequence was flanked by Bst BI and Xho I restriction sites for cloning purposes; (2) Fragment B: *loxPint* followed by the 3D7 *msp1$_{19}$* sequence and endogenous 3D7 *msp1* 3′ UTR. The sequence encoding the SUB2 cleavage site (Leu-Asn) in the 3D7 *msp1* sequence was mutated to encode Pro-Pro. The synthetic fragment was flanked by Not I and Nde I restriction sites; (3) Fragment C: *loxPint* followed by 3D7 *msp1$_{19}$* sequence. A chimeric sequence comprising 3D7 target, Wellcome *msp1$_{19}$* and PbDT3′ UTR was excised from pHH1-wMSP1-wt (**Child et al., 2010**) using Hpa I and Not I and cloned into pGEM-T Easy pre-digested with Sph I (T4 polymerase blunt ended) and Not I, generating plasmid pGEM-MSP1. Fragment A was cloned into pGEM-MSP1 using Xho I and Bst BI giving rise to plasmid pGEM-MSP1(A). Plasmid pGEM-MSP1$_{PP}$ was generated by cloning B into pGEM-MSP1(A) using Not I and Nde I restriction digestion. Control plasmid pGEM-T MSP1$_{LN}$ was obtained by cloning fragment C into plasmid pGEM-MSP1$_{PP}$ using Not I and Pst I. All PCR screening relied on forward primer MSP1_Int_F in combination with MSP1_EndCont_R for the endogenous locus (amplicon: 1.4 kb) and wMSP119_Int_R for integrated (amplicon: 1.4 kb) and excised (amplicon: 1.4 kb) loci.

## Construct design to generate parasites conditionally expressing AMA1 refractory to cleavage by SUB2 and ROM4 (iΔRΔS)

A synthetic sequence was ordered comprising a small section of the final 3D7 *ama1* target region, followed by recodonised segment of the *ama1* 3′ from *P. falciparum* strain FVO (**Kocken et al., 2002**) flanked by *loxPS2int* sequence, 3D7 *ama1* 3′ and finally native 3D7 *ama1* 3′ UTR (GeneArt). The recodonised FVO sequence was modified by amino acid replacement to include a haemagglutinin (HA) epitope tag within the stub region between domain III and the transmembrane region of AMA1 (IPEHKPTYD/YPYDVPDYA) as previously described (**Collins et al., 2007**). The native *ama1* 3′ sequence was altered to introduce the TS/PP mutations at the P1-P1′ positions of the SUB2 cleavage site, as well as an HA epitope tag. This HA tag was introduced by amino acid replacement into an unstructured loop in domain III (KRIKLNDND/YPYDVPDYA) as previously described (**Collins et al., 2009**). Native 3D7 *ama1* target sequence was amplified from 3D7 genomic DNA using primers endo_SacII_F and endo_dsBclI_R and cloned into pGEM-T using SacII and NdeI, giving rise to plasmid pGEM-pfama1. The GeneArt generated synthetic sequence was cloned into pGEM-pfama1 using Sac II and Nde I generating plasmid pGEM-A1-3′UTR. The PbDT3′ sequence was excised from pHH1-5′sgPfa1HA (**Collins et al., 2007**) and inserted between the FVO *ama1* sequence and the second *loxPS2int* using Avr II and Not I giving rise to pGEM-T-iΔS_A1.

The region encoding the SUB2 cleavage site was amplified from plasmid pGEM-T-iΔS_A1 using primer pairs endo_usPacI_F/A1_S2_C_R and A1_S2_C_F/A1_dsBsu361_R. Overlapping PCR was carried out using the resulting fragments and primers endo_usPacI_F and A1_dsBsu361_R. The resulting PCR product was cloned into plasmid pGEM-T-iΔS_A1 using restriction enzymes PacI and Bsu361 giving rise to the control plasmid pGEM-T- iA1_C.

Mutations at the P1 position of the rhomboid cleavage site (Y550A) in the region encoding the AMA1 TMD were introduced by overlapping PCR. Briefly, PCR fragments were obtained from pGEM-T-iΔS_A1 using primer pairs endo_usPac1/A1_A550_Y_R and A1_A550_Y_F/ama1 3′_dsNde1. Overlapping PCR was carried out using primers endo_usPac1 and ama1 3′_dsNde1. This product was cloned into pGEM-T- iA1_C and pGEM-T-iΔS_A1 giving rise to plasmids pGEM-T-iΔR_A1 and pGEM-T-iΔRΔS_A1. PCR screening relied on use of the forward primer A1_Int_F1 in combination with reverse primers A1_EndCont_R1 for the endogenous locus (amplicon; 1.3 kb), A1_Int_R for the integrated locus (1.4 kb) and A1_dsBsu361_R for both non-excised (3.9 kb) and excised (1.9 kb) loci.

## Construct design for direct insertion of SUB2/ROM4 cleavage site mutations in the ama1 gene (line dΔRΔS)

Two gBlocks were ordered (IDT) comprising recodonised *ama1* sequence from upstream of the Avr II site and incorporating a 3′ Bsu361 site at the position equivalent to that in the endogenous 3D7 sequence. The control gBlock (C) had no additional modifications whilst the other had mutations at

the SUB2 and rhomboid cleavage sites described above (ΔRΔS). The gBlocks were cloned into pCR-Blunt (Thermo Fisher Scientific) and then excised using Avr II and Bsu361 and cloned into pGEM-T-iA1-C pre-digested with the same restriction enzymes giving rise to plasmids pGEM-T_dA1_C and pGEM-T_dA1_ΔRΔS. PCR screening was the same as for the conditionally uncleavable AMA1 lines (iΔRΔS_A1).

## Construct for generation of parasites for conditional disruption of AMA1 expression

The mCherry sequence was amplified from plasmid pDC2-mCherry-MCS using primers mCherry_PacI_F and mCherry_Bsu361_R. The resulting fragment was digested with Pac I and Bsu361 and cloned into pGEM-T-iA1-C pre-digested with the same enzymes, giving rise to plasmid pGEM-T-ΔA1. PCR screening relied on use of the forward primer A1_Int_F1 in combination with reverse primers A1_EndCont_R1 for the endogenous locus (1.3 kb), A1_Int_R for the integrated locus (1.4 kb) and A1_dsBsu361_R for both non-excised (3.8 kb) and excised (2.0 kb) loci.

### Guide/Cas9 plasmids

Guide RNA sequences specific for each target gene of interest were identified using Protospacer (https://sourceforge.net/projects/protospacerwb/) or Benchling (www.benchling.com). Selected guide RNA sequences were cloned into plasmid pDC2-Cas9-hDHFRyFCU, containing a Cas9 expression cassette and the drug selection marker human dihydrofolate reductase (*hdhfr*) as previously described (*Knuepfer et al., 2017*). Complementary oligonucleotides were designed that upon annealing generated sticky ends, compatible with the ends resulting from Bbs1 digestion of plasmid pDC2-Cas9-hDHFRyFCU. To generate conditional-uncleavable forms of AMA1, the *ama1* gene was targeted with guides A1_G1 (oligonucleotides A1-G1_F and A1_G1_R) and A1_g3 (oligonucleotides A1_G3_F and A1_G3_R) which gave rise to plasmids pDC_A1G1 and pDC_A1G3, respectively when cloned into pDC2-Cas9-hDHFRyFCU. Targeting of the *msp1* gene to generate conditional-uncleavable MSP1 was carried out using the *msp1* guide (oligonucleotides msp1guide_F and msp1guide_R) resulting in plasmid pDC_M1G. To generate a conditional knock-out AMA1 line, AMA1guide2 (oligonucleotides AMA1guide2_ and AMA1guide2_R) were cloned into pDC2-Cas9-hDHFRyFCU giving rise to plasmid pDC_A1G2. The U6 cassette from plasmid pDC_A1G2 was amplified using Q5 high fidelity DNA Polymerase (New England BioLabs; NEB) with primers HD104_Cas9UTR_F and HD104_Cas9UTR_R. The resulting PCR product was cloned into pDC_A1G3 pre-digested with Sal I using Gibson Assembly master mix (NEB) according to the manufacturer's instructions giving rise to plasmid pDC_A1G2/3.

## Oligonucleotide primers used in this study

| Primer name: | Sequence: |
|---|---|
| sgS2_HBA_F | GGACACCACGTTAACAGTACCATTCCGGACAACTCATACAAAATCTTCACCGG |
| sgS2_X1_R | CGTAAGGGTACTCGAGCTTCATGAACATGTCGTCCAGCTGGTTCATTGCC |
| S2endo_HpaI_F | CAAAATGGTTAACGGAAAATACAAACCTTTATGATGGAACGGGG |
| S2endo_BspEI_R | GGGTATAAAGGTGAAATATCATTATCGTTACTTTTATTTCCGG |
| endo_SacII_F | CGAATATCCATTACCGCGGGAAC |
| endo_dsBclI_R | CTATCTGCATATGAAGCACCAGTGGGAAG |
| endo_usPac1_F | CATGGTAAGGGTTATAATTGGGG |
| A1_S2_C_R | CTACAACTTCATTATTTGATGTTACTTCTGCCC |
| A1_S2_C_F | GGGCAGAAGTAACATCAAATAATGAAGTTGTAG |
| ama1 3'_dsNde1 | GCTTTAACTTTAAAGTTACAACATC |
| A1_A550_Y_R | GACAGCAGCTGATGAATAAATTATAATTTTC |
| A1_A550_Y_F | GAAAATTATAATTTATTCATCAGCTGCTGTC |
| mCherry_PacI_F | CCAACATGTTTAATTAACATGAAGGTGAGCAAGGGC |

*Continued on next page*

*Continued*

| Primer name: | Sequence: |
| --- | --- |
| mCherry_Bsu361_R | AAGATGCCTCAGGTTACTTGTACAGCTCGTCCATGCCGCC |
| A1-G1_F | ATTGTCCTGAGGCATCTTTTTGGG |
| A1_G1_R | AAACCCCAAAAAGATGCCTCAGGA |
| AMA1_G3_F | ATTGTTTATATCTATCTGCTTTAA |
| AMA1_G3_R | AAACTTAAAGCAGATAGATATAAA |
| msp1guide_F | ATTGTGAAATGTTTAACATATCT |
| msp1guide_R | AAACAGATATGTTAAACATTTCA |
| AMA1guide2_F | ATTGATTTTCATTTTATCATAAGT |
| AMA1guide2_R | AAACACTTATGATAAAATGAAAAT |
| HD104_Cas9UTR_F | ATCAAATAGCATGCCTGCAGGTCGACGACATTTGGATTTCTACACATCTTG |
| HD105_Cas9UTR_R | CAAATGTCGGATCCTCTAGAGTCGACGGATCCGCCTTAAAAACTTC |
| S2assplasfor2 (green) | ACTCAAGATTTTTATAACTTCATGG |
| pHH1_R1 (pink) | CCGCTCGCCGCAGCCGAACGACCG |
| SUB2for_Int (black) | CTCATCAGAGATATATAGTACTATCAAACAATGG |
| SUB2rev_wt (blue) | TAGAATATATGTGATGACCTGGGGC |
| SUB2rev_Int (red) | GCCAGGATGTGAACAGACTTGTGACC |
| nonexSUB2_F | GTAGTGTGGCCTCCATTAGCAG |
| A1seq1for | GGAACTCAATATAGACTTCCATCAGGG |
| A1seq6 rev | CAACTTCGATGGGATGGGACAAAGCAG |
| sgSUB2_5F | AAAGCTTCCTGTCTGACCTGGAACAGAACTACAAGCCCCT |
| sgSUB2_31R | GTAAATACCGTCCATGTCCATGAACTTAGGGTTTTTGGTC |
| AMA1_int_F1 (a) | GTGATGTGTATCGTCCAATC |
| AMA1_int_R (c) | GGATGAGACAGGGCAGTAGTCGC |
| A1_EndCont_R1 (b) | CATCATTATCATTTAATTTAATTCG |
| A1_dsBsu361_R (d) | GTGATGCTCTTTTTTCTTCGCCCC |
| MSP1_Int_F1 (e) | GTAACTCCACCTCAACCAGATG |
| wMSP1$_{19}$_Int_R (g) | CGCATTTATCGCCTTCCTG |
| MSP1_EndCont_R (f) | CATGTGGCATCTGCATCACATCCACC |

(Colours and letters in parenthesis in the left-hand column correspond to primer annotations in *Figure 1*, *Figure 4*, and *Figure 6—figure supplement 1*, *Figure 7—figure supplement 1* and *Figure 7—figure supplement 2*).

## Generation of transgenic *P. falciparum* lines

Transgenic parasite lines were generated on the background of either the 1G5DC (*Collins et al., 2013a*) or B11 DiCre-expressing parasite lines. In all cases DNA was introduced by electroporation of purified schizonts with sterile DNA, in Amaxa Primary cell solution P3, using a 4D-Nucleofector (Lonza). For single cross-over homologous recombination, 80 µg of purified DNA was used. Drug pressure (WR99210, 2.5 nM; Sigma-Aldrich) was applied 24 hr post transfection and maintained until viable parasites were obtained. For stable transgenic lines, parasites were subjected to multiple rounds of drug pressure as previously described (*Harris et al., 2005*). For Cas9-targeted double cross-over homologous recombination, 20 µg of the guide/Cas9 plasmid and 60 µg of *Sap* I or *Nar* I linearised repair plasmid were used (*Knuepfer et al., 2017*). Drug selection pressure (2.5 nM WR99210) was applied 24 hr following transfection and maintained for 4 d before removal and continued culture of the parasites.

## Plaque assay and limiting dilution cloning of transgenic parasite lines

Plaque assays of transgenic parasites were as described previously (*Thomas et al., 2016*). For limiting dilution cloning, 200 µl of culture containing parasites at an estimated 1 parasite per 200 µl and 0.75% haematocrit, was added to each well of a flat-bottomed 96-well plate. Wells containing single plaques were identified after 10–14 days using an inverted microscope and the parasites expanded for further analysis.

## PCR validation of transgenic parasite lines

Parasite pellets were lysed with 0.15% saponin and genomic DNA extracted using a Qiagen DNeasy Blood and Tissue Kit (Qiagen). PCR screens were carried out using GoTaq Green (Promega) with the primer combinations described.

## Invasion and growth assays

For estimation of invasion efficiency, Percoll-enriched schizonts were added to fresh RBCs to obtain a parasitaemia of 5–10% in 4 ml at a 2% haematocrit in RPMI-Albumax. Cultures were incubated at 37°C in a shaking incubator for 4 hr. Ring-stage parasites were purified as described above and returned to culture in RPMI-Albumax. Samples were taken, fixed with 0.2% glutaraldehyde and stored at 4°C for flow cytometry analysis. The remaining culture was followed for 48 hr to check parasite development. Giemsa-stained thin films were prepared as required for microscopic analysis.

For longer-term replication assays, cultures were synchronised as described and resulting ring-stage cultures maintained for 24 hr to mature to trophozoite stage. Parasitaemia levels were calculated and cultures adjusted to 0.1% parasitaemia, 2% haematocrit in a final volume of 2 ml per well of a six well plate. Samples were then taken at t = 0, 48, 96, and 144 hr, fixed in 0.2% glutaraldehyde and stored at 4°C for flow cytometry analysis. Culture media were replaced at 96 hr and 120 hr.

## Flow cytometry for parasite quantification

Parasite samples were fixed in 0.2% glutaraldehyde in PBS and stored at 4°C. Cells were prepared for analysis by staining with 2 x SYBR Green I nucleic acid gel stain (Life Technologies) for 30 min at 37°C. Labelling was stopped with an equal volume of PBS and samples analysed using a BD Fortessa flow cytometer (BD Biosciences) with BD FACSDiva software (BD Biosciences). Total RBC numbers were calculated using forward- and side-scatter whilst fluorescence was detected using the 530/30 blue detection laser. Fluorescence intensity was used to distinguish uninfected from infected RBCs, low fluorescence indicating uninfected cells and gating fixed accordingly. Data were analysed using FlowJo.

## Immunofluorescence analysis (IFA)

Thin films of parasite cultures were made on glass slides, then air dried and stored under dessicant at −80°C. Slides were thawed, fixed in 4% paraformaldehyde for 30 min then permeabilized in 0.1% Triton X-100 in phosphate buffered saline (PBS) for 10 min prior to blocking in 3% (w/v) BSA in PBS overnight. Antibody incubations were carried out for 30 min at 37°C in a humidified chamber followed by washing twice for 5 min each in PBS. All antibodies were diluted into 3% (w/v) BSA in PBS. For microscopic imaging, samples were mounted in Vectashield containing DAPI (Vector laboratories). Images were acquired using a Nikon Eclipse Ni microscope with a 100x Plan Apo λ HA 1.45 objective, with a Hamamatsu C11440 digital camera and processed using Fiji.

## Visualisation of RBC ghost formation using fluorescent phalloidin

Synchronous ring-stage SUB2HA3:loxP parasites were mock -or RAP-treated, then ~44 hr later schizonts were Percoll-enriched and added to fresh RBCs for 4 hr to allow egress and invasion. Without an intervening centrifugation step, a sample of the culture was supplemented with Hoechst 33342 (2 µg/ml; ThermoFisher) and Alexafluor 488 Phalloidin (diluted 1:50; ThermoFisher) and immediately applied to a viewing chamber for live imaging as described previously (*Collins et al., 2013b*). Z-stack images were acquired at 2 µm intervals using a Nikon Eclipse Ni microscope with a 100x Plan Apo λ HA 1.45 objective and a Hamamatsu C11440 digital camera. The number of phalloidin-labelled ghosts per image was counted manually for each treatment.

## Haemoglobin release assay

DMSO- or RAP-treated mature SUB2HA3:loxP schizonts were Percoll-enriched and resuspended in fresh RPMI 1640-Albumax. Schizont suspensions were mixed with an equal volume of either RPMI 1640-Albumax (control; schizonts only) or RBCs at a 2% haematocrit in RPMI 1640-Albumax (co-culture; schizonts plus RBCs). Two additional control samples were set up comprising a similar volume of only RBCs (2% haematocrit) in RPMI 1640-Albumax mixed with an equal volume of RPMI 1640-Albumax (RBC controls). RBCs from one of these samples were immediately pelleted by centrifugation, the supernatant removed and 200 μl water added to obtain hypotonic lysis of the RBCs. Complete lysis was ensured by subjecting the sample to repeated freeze thawing. RPMI 1640-Albumax was then added to return the sample to the original volume. This sample acted as total RBC lysate control.

The schizont only control samples, the co-culture samples and the second RBC control sample were incubated with gentle shaking for 4–6 hr at 37°C to allow egress and/or invasion. In some experiments the medium was supplemented with cytochalasin D (10 μM) or 0.5% DMSO (vehicle only) as a control. All samples were then clarified by centrifugation and the culture medium filtered through a 2 μm filter. The samples of culture medium were then subjected to serial 2-fold dilutions in RPMI 1640-Albumax, and the haemoglobin content relative to that of the total RBC lysate determined by absorbance at 405 nm (*Snell and Marini, 1988*) using a Spectramax M multi-mode microplate reader (Molecular Devices). The pelleted cells from the schizont-containing samples were centrifuged over Percoll cushions to remove residual schizonts and the newly-invaded rings returned to culture. Samples of these were taken immediately as well as 18 hr and 24 hr later for flow cytometry and microscopic analysis.

## Time-lapse DIC video microscopy of egress and invasion

Visualisation of egress was as described previously (*Thomas et al., 2018*). Briefly, mature schizonts were Percoll-enriched then returned to culture for 4 hr in RPMI 1640-Albumax supplemented with compound 2 (4-[7-[(dimethylamino)methyl]−2-(4-fluorphenyl)imidazo[1,2-$\alpha$]pyridin-3-yl]pyrimidin-2-amine, C2; 1 μM) to arrest and synchronise the schizonts in a highly mature, pre-egress stage. The parasites were then pelleted, washed twice in pre-warmed, gassed RPMI 1640-Albumax medium without C2, then finally pelleted at 1800 x g for 1 min and immediately resuspended at $\sim 1 \times 10^7$ cells/μl in pre-warmed gassed medium without C2. The parasites were introduced by capillary flow into a pre-warmed viewing chamber, made by adhering a 22 × 64 mm borosilicate glass coverslip to a microscope slide (*Collins et al., 2013b*), then at once imaged on a 37°C heated stage on a Zeiss Axio Imager M1 microscope, using a EC Plan-Neofluar 1006/1.3 oil immersion DIC objective fitted with an AxioCam MRm camera. Images were collected at 5 s intervals for 30 mins, annotated and exported as QuickTime movies using Axiovision 3.1.

To image invasion, Percoll-enriched, highly synchronous mature schizonts from RAP- or DMSO-treated cultures were added to fresh RBCs at a 2% haematocrit in pre-warmed, gassed RPMI 1640-Albumax to achieve a parasitaemia of ~10%. The cells were introduced into a pre-warmed viewing chamber, transferred to the heated stage (maintained at 37°C) of a Nikon Eclipse Ni-E wide field microscope and imaged with a Hamamatsu C11440 digital camera using a Nikon N Plan Apo 100 × 1.45 NA oil immersion objective. DIC images were acquired at 1 s intervals for 30 min, exported as ND2 files and annotated using Fiji (Image J version 2).

## Proteomic analysis

Supernatants of DMSO-and RAP-treated SUB2HA3:loxP schizonts samples following egress and invasion in Albumax-free RPMI 1640 medium were subjected to SDS-PAGE on 16.5% Mini-PROTEIN Tris-Tricine gels (BioRad). Reduced and alkylated proteins from band slices excised from the entire separation profile were in-gel digested at 37°C overnight with 100 ng trypsin (modified sequencing grade, Promega). Supernatants were dried in a vacuum centrifuge and resuspended in 0.1% TFA. Peptides were loaded onto an Ultimate 3000 nanoRSLC HPLC (Thermo Scientific) coupled to a 2 mm x 0.3 mm Acclaim Pepmap C18 trap column (Thermo Scientific) at 15 μl/min prior to elution at 0.25 μl/min through a 50 cm x 75 μm EasySpray C18 column into an Orbitrap Fusion Lumos Tribrid Mass Spectrometer (Thermo Scientific). A gradient of 2–32% acetonitrile in 0.1% formic acid over 45 min was used, prior to washing (80% acetonitrile, 0.1% formic acid) and re-equilibration. The

Orbitrap was operated in data-dependent acquisition mode with a survey scan at a resolution of 120,000 from m/z 300–1500, followed by MS/MS in TopS mode. Dynamic exclusion was used with a time window of 20 s. The Orbitrap charge capacity was set to a maximum of $1 \times e^6$ ions in 10 ms, whilst the LTQ was set to $1 \times e^4$ ions in 100 ms. Raw files were processed using Maxquant 1.3.0.5 and Perseus 1.4.0.11. A decoy database of reversed sequences was used to filter false positives, at a peptide false detection rate of 1%.

## Serial block-face scanning EM (SBF-SEM) and transmission EM (TEM)

Mature schizonts enriched from DMSO- and RAP-treated SUB2HA3:loxP cultures were incubated with fresh RBCs for 4 hr to allow invasion, then newly-invaded rings isolated as described. Either immediately or following further culture for 20 hr, the parasites were washed, fixed with 2.5% glutaraldehyde 4% formaldehyde in 0.1 M phosphate buffer (PB), then further washed in 0.2 M phosphate buffer (PB), embedded in 4% low gelling temperature agarose and cut into 1 mm$^3$ blocks (*Hanssen et al., 2010*).

For TEM, sample blocks were washed $4 \times 15$ min in PB and post-fixed in 1% osmium tetroxide/1.5% potassium ferrocyanide for 1 hr at 4℃. After further washes in PB at room temperature, the blocks were stained in 1% tannic acid for 45 min and quenched in 1% sodium sulphate for 5 min. The blocks were then washed $4 \times 5$ min in distilled water and dehydrated through an ethanol series (70–100%, $2 \times 10$ min each). Finally, the sample blocks were embedded by incubating in propylene oxide (PO) for 10 min, then overnight in a 1:1 PO:Epon 812 (TAAB) resin mixture, followed by 3 changes of pure Epon resin over 2 d, and baking at 60℃ for 24 hr. For TEM image collection, 80 nm sections were stained with lead citrate and imaged in a Tecnai Spirit BioTwin (Thermofisher Scientific) transmission electron microscope.

For SBF-SEM, samples were embedded in agarose as above, then processed using a protocol adapted from the NCMIR protocol (https://ncmir.ucsd.edu/sbem-protocol). Briefly, the sample blocks were washed $4 \times 15$ min in 0.1 M PB and post-fixed in 2% osmium tetroxide/1.5% potassium ferrocyanide for 1 hr at 4℃, washed $5 \times 3$ min in water (also after the following steps up to the dehydration), incubated in 1% w/v thiocarbohydrazide for 20 min before a second staining with 2% osmium tetroxide for 30 min, followed by incubation overnight in 1% aqueous uranyl acetate at 4℃. The sample blocks were then stained with Walton's lead aspartate for 30 min at 60℃ and dehydrated through an ethanol series on ice (25–100%, 10 min each). Finally, the sample blocks were embedded as above, except Durcupan (TAAB) resin was used and the baking step was for 48 hr. Embedded samples were trimmed and mounted on pins using conductive epoxy glue (*Russell et al., 2017*). SBF-SEM images were collected using a 3View 2XP system (Gatan Inc), mounted on a Sigma VP scanning electron microscope (Zeiss). Images were collected at 1.9 kV, with a 30 μm aperture, at 6 Pa chamber pressure, and a 2 μs dwell time. The datasets were $69.63 \times 69.63 \times 13$ μm, and $69.63 \times 69.63 \times 12.6$ μm in xyz, consisting of 260 and 252 serial images of 50 nm thickness and a pixel size of $8.5 \times 8.5$ nm. The total volume of the datasets was ~63,028 μm$^3$ and ~61,089 μm$^3$. SBF-SEM movies were processed in Fiji to adjust brightness and contrast, binned to $4096 \times 4,096$ pixels, and then converted to AVI, before exporting from Quicktime Pro as $950 \times 950$ pixel H.264 movies.

## Quantification and statistical analysis

All graphs of experimental data and statistical analysis were generated using GraphPad Prism 8.0. Statistical analysis methods for each experiment are stated in the corresponding figure legend.

## Lead contact and materials availability

Further information and request for resources and reagents should be directed to and will be fulfilled by corresponding contact Michael J. Blackman (mike.blackman@crick.ac.uk). All new plasmid and parasite lines generated in this study can be requested from the corresponding author and will be provided subject to a completed Material Transfer Agreement.

## Acknowledgements

This work was supported by funding to MJB from the Francis Crick Institute (https://www.crick.ac.uk/) which receives its core funding from Cancer Research UK (FC001043; https://www.cancerresearchuk.org), the UK Medical Research Council (FC001043; https://www.mrc.ac.uk/), and the

Wellcome Trust (FC001043; https://wellcome.ac.uk/). The work was also supported by Wellcome ISSF2 funding to the London School of Hygiene & Tropical Medicine.

## Additional information

### Funding

| Funder | Grant reference number | Author |
|---|---|---|
| Cancer Research UK | FC001043 | Christine R Collins<br>Fiona Hackett<br>Steven A Howell<br>Ambrosius P Snijders<br>Matthew Robert Geoffrey Russell<br>Lucy Collinson<br>Michael J Blackman |
| Medical Research Council | FC001043 | Christine R Collins<br>Fiona Hackett<br>Steven A Howell<br>Ambrosius P Snijders<br>Matthew Robert Geoffrey Russell<br>Lucy Collinson<br>Michael J Blackman |
| Wellcome Trust | FC001043 | Christine R Collins<br>Fiona Hackett<br>Steven A Howell<br>Ambrosius P Snijders<br>Matthew Robert Geoffrey Russell<br>Lucy Collinson<br>Michael J Blackman |
| Wellcome Trust | ISSF2 | Michael J Blackman |

The funders had no role in study design, data collection and interpretation, or the decision to submit the work for publication.

### Author contributions

Christine R Collins, Conceptualization, Formal analysis, Investigation, Methodology, Writing - original draft, Writing - review and editing; Fiona Hackett, Conceptualization, Formal analysis, Investigation, Methodology, Writing - original draft; Steven A Howell, Data curation, Formal analysis, Investigation, Methodology; Ambrosius P Snijders, Conceptualization, Data curation, Formal analysis, Supervision, Funding acquisition, Investigation, Methodology, Writing - original draft; Matthew RG Russell, Formal analysis, Investigation, Visualization, Methodology, Writing - original draft; Lucy M Collinson, Data curation, Supervision, Funding acquisition, Methodology, Writing - original draft, Project administration; Michael J Blackman, Conceptualization, Formal analysis, Supervision, Funding acquisition, Investigation, Writing - original draft, Project administration, Writing - review and editing

### Author ORCIDs

Christine R Collins  https://orcid.org/0000-0001-5191-7634
Matthew RG Russell  https://orcid.org/0000-0003-4608-7669
Michael J Blackman  https://orcid.org/0000-0002-7442-3810

### Decision letter and Author response

Decision letter https://doi.org/10.7554/eLife.61121.sa1
Author response https://doi.org/10.7554/eLife.61121.sa2

## Additional files

### Supplementary files

• Transparent reporting form

## Data availability

All data generated or analysed during this study are included in the manuscript and supporting files. Source data files have been provided for Figure 3. In addition, the raw mass spectrometry proteomics data are available via ProteomeXchange with the dataset identifier PXD021843.

The following dataset was generated:

| Author(s) | Year | Dataset title | Dataset URL | Database and Identifier |
|---|---|---|---|---|
| Howell SA, Snijders AP | 2020 | The malaria parasite sheddase SUB2 governs host red blood cell membrane sealing at invasion | https://www.ebi.ac.uk/pride/archive/projects/PXD021843 | PRIDE, PXD021843 |

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

# Appendix 1

**Appendix 1—key resources table**

| Reagent type (species) or resource | Designation | Source or reference | Identifiers | Additional information |
|---|---|---|---|---|
| gene (*Plasmodium falciparum*) | *pfsub2* | PlasmoDB ([https://plasmodb.org](https://plasmodb.org)) | PF3D7_1136900 | *P. falciparum* SUB2 gene |
| genetic reagent (*P. falciparum*) | SUB2HA3:loxP | This paper | | For inducible disruption of PfSUB2. Line maintained in and available from Blackman lab, Francis Crick Institute. |
| genetic reagent (*P. falciparum*) | iMSP1$_{PP}$ | This paper | | For inducible mutagenesis of MSP1. Line maintained in and available from Blackman lab, Francis Crick Institute. |
| genetic reagent (*P. falciparum*) | iMSP1$_{LN}$ | This paper | | Control for inducible allelic replacement of MSP1. Line maintained in and available from Blackman lab, Francis Crick Institute. |
| genetic reagent (*P. falciparum*) | dΔRΔS | This paper | | For constitutive expression of cleavage-resistant AMA1. Line maintained in and available from Blackman lab, Francis Crick Institute. |
| genetic reagent (*P. falciparum*) | iΔRΔS | This paper | | For inducible mutagenesis of AMA1. Line maintained in and available from Blackman lab, Francis Crick Institute. |
| genetic reagent (*P. falciparum*) | iAMA1_C | This paper | | Control for inducible allelic replacement of AMA1. Line maintained in and available from Blackman lab, Francis Crick Institute. |
| cell line (*P. falciparum*) | B11 | *Perrin et al., 2018* (PMID:29970464) | | DiCre-expressing parasite line. Maintained in and available from Blackman lab, Francis Crick Institute |
| cell line (*P. falciparum*) | 1G5DC | *Collins et al., 2013a* (PMID:23489321) | | DiCre-expressing parasite line. Maintained in Blackman lab, Francis Crick Institute |
| transfected construct (*P. falciparum*) | pSUB2-BSD | This paper | | For ectopic expression of SUB2. Available from Blackman lab. |
| transfected construct (*P. falciparum*) | pGEM-MSP1$_{PP}$ | This paper | | For inducible disruption of SUB2-mediated MSP1 shedding. Available from Blackman lab. |
| transfected construct (*P. falciparum*) | pGEM-MSP1$_{LN}$ | This paper | | For generation of inducible control parasite line. Available from Blackman lab. |
| transfected construct (*P. falciparum*) | pGEM-T-iΔRΔS_A1 | This paper | | For inducible disruption of SUB2-mediated AMA1 shedding. Available from Blackman lab. |

*Continued on next page*

*Appendix 1—key resources table continued*

| Reagent type (species) or resource | Designation | Source or reference | Identifiers | Additional information |
|---|---|---|---|---|
| transfected construct (*P. falciparum*) | pGEM-T- iA1_C | This paper | | For generation of inducible control parasite line. Available from Blackman lab. |
| transfected construct (*P. falciparum*) | pGEM-T_dA1_ΔRΔS | This paper | | For direct disruption of SUB2-mediated AMA1 shedding. Available from Blackman lab. |
| transfected construct (*P. falciparum*) | pGEM-T_dA1_C | This paper | | To generate control parasite line. Available from Blackman lab. |
| transfected construct (*P. falciparum*) | pDC2-mCherry-MCS | This paper | Mapping the mouse Allelome reveals tissue-specific regulation of allelic expression | Control vector for pSUB2-BSD. Available from Blackman lab. |
| biological sample (*Homo sapiens*) | Human red blood cells | UK NHS Blood and Transplant | | Provided anonymised. |
| antibody | X509 anti-MSP1 (human monoclonal) | *Blackman et al., 1991* (PMID:1723148) | | (1 µg/mL) |
| antibody | Anti-AMA1 (rabbit polyclonal) | *Collins et al., 2009* (PMID:19165323) | | Immunofluorescence (IFA) (1:500), western blot (1:1000) |
| antibody | Anti-SERA5 (rabbit polyclonal) | *Stallmach et al., 2015* (PMID:25599609) | | IFA (1:500), western blot (1:1000) |
| antibody | 3F10 High affinity anti-HA (rat monoclonal) | Roche | Cat# 11867423001, RRID:AB_390918 | IFA (1:500), western blot (1:1000) |
| antibody | Anti-EBA175 (rabbit polyclonal) | *Withers-Martinez et al., 2008* (PMID:18036613) | | IFA (1:500) |
| antibody | 24C6 anti-RON4 (mouse monoclonal) | *Roger et al., 1988* (PMID:3278223) | | IFA (1 µg/mL) |
| antibody | 89.1 anti-MSP1$_{83}$ (mouse monoclonal) | *Holder and Freeman, 1982* (PMID:6752328) | | IFA, western blot (1 µg/mL) |
| antibody | 2F10 anti-MSP1$_{19}$ (mouse monoclonal) | *Burghaus and Holder, 1994* (PMID:8078519) | | IFA (1 µg/mL) |
| antibody | 111.4 anti-MSP1$_{19}$ Wellcome-type specific (mouse monoclonal) | *Holder et al., 1985* (PMID:2995820) | | IFA (1 µg/mL) |
| antibody | Anti-PTRAMP (rabbit polyclonal) | *Green et al., 2006* (PMID:16879884) | | Western blot (1:1000) |
| antibody | AlexaFluor 594 conjugated anti-rabbit; (goat polyclonal) | ThermoFisher | Cat# A32740, RRID: AB_2762824 | IFA (1:10,000) |
| antibody | AlexaFluor 488 conjugated anti-rabbit; (goat polyclonal) | ThermoFisher | Cat# A-11008, RRID: AB_143165 | IFA (1:10,000) |

*Continued on next page*

*Appendix 1—key resources table continued*

| Reagent type (species) or resource | Designation | Source or reference | Identifiers | Additional information |
|---|---|---|---|---|
| antibody | AlexaFluor 594 conjugated anti-mouse; (goat polyclonal) | ThermoFisher | Cat# A-11032, RRID: AB_2534091 | IFA (1:10,000) |
| antibody | AlexaFluor 488 conjugated anti-mouse; (goat polyclonal) | ThermoFisher | Cat# A-11029, RRID: AB_138404 | IFA (1:10,000) |
| antibody | AlexaFluor 594 conjugated anti-human; (goat polyclonal) | ThermoFisher | Cat# A-11014, RRID: AB_2534081 | IFA (1:10,000) |
| antibody | AlexaFluor 488 conjugated anti-human (goat polyclonal) | ThermoFisher | Cat# A-11013, RRID: AB_2534080 | IFA (1:10,000) |
| antibody | Biotinylated anti-rat (goat polyclonal) | Sigma-Aldrich | Cat# AP183B, RRID: AB_92595 | IFA (1:1000) |
| sequence-based reagent | All primers and guides: see primer table | This paper | | Used for diagnostic PCR. Available from Blackman lab. |
| peptide, recombinant protein | AlexaFluor 488 Phalloidin | ThermoFisher | Cat# A12379 | |
| commercial assay or kit | Gibson Assembly master mix | NEB | Cat# E2611L | |
| commercial assay or kit | pCR-Blunt II-TOPO vector | ThermoFisher | Cat# 280002 | |
| commercial assay or kit | pGEM-T Easy vector system | Promega | Cat# A1360 | |
| chemical compound, drug | SYBR Green I | ThermoFisher | Cat# S7563 | |
| chemical compound, drug | Streptavidin peroxidase | Sigma-Aldrich | Cat# S2438 | |
| chemical compound, drug | AlexaFluor 594 conjugated Streptavidin | ThermoFisher | Cat# S32356 | |
| chemical compound, drug | AlexaFluor 488 conjugated Streptavidin | ThermoFisher | Cat# S32354 | |
| chemical compound, drug | WR99210 | Sigma-Aldrich | Cat# W1770 | Antifolate drug. The *hdhfr* gene confers resistance. |
| chemical compound, drug | Rapamycin | Sigma-Aldrich | Cat# R0395-1MG | |
| chemical compound, drug | Cytochalasin D | Sigma-Aldrich | Cat# C8273 | |
| chemical compound, drug | Compound 2 (4-[7-dimethylamino)methyl]—2-(4-fluorphenyl)imidazo[1,2-α]pyridine-[3-yl]pyrimidin-2-amine | LifeArc (https://www.lifearc.org/) | | Kindly provided by Dr Simon A Osborne (LifeArc) |

*Appendix 1—key resources table continued*

| Reagent type (species) or resource | Designation | Source or reference | Identifiers | Additional information |
|---|---|---|---|---|
| software, algorithm | Protospacer | Cameron MacPherson, BIHP, Institut Pasteur, Paris, France | https://sourceforge.net/projects/protospacerwb/ | |
| software, algorithm | FlowJo for Mac (version 10.3.0) | Becton Dickinson Life Sciences | RRID:SCR_008520 | |
| software, algorithm | GraphPad Prism 8.0 | https://www.graphpad.com/ | RRID:SCR_002798 | |
| software, algorithm | Fiji (Image J version 2.0) | Imagej.net | RRID:SCR_003070 | |
| software, algorithm | BD FACSDiva software | BD Bioscience | RRID:SCR_001456 | |
| software, algorithm | Axiovision 3.1 | Zeiss | RRID:SCR_002677 | |
| software, algorithm | MaxQuant 1.3.0.5 | Max Planck Institute of Biochemistry | RRID:SCR_014485; https://www.maxquant.org/ | |
| software, algorithm | Perseus 1.4.0.11 | Max Planck Institute of Biochemistry | RRID:SCR_015753; http://www.perseus-framework.org | |

