## [Decision Letter]

**Acceptance summary:**

The process of host cell invasion is an essential step in the life cycle of apicomplexan parasites, and involves many cell biological processes unique to these organisms. Central to these events is the interaction between parasite adhesins and the host cell membrane. The present study uses elegant genetics and a collection of analytical approaches to investigate the function of a parasite protease, SUB2, previously shown to mediate proteolysis of several key adhesins. The authors conclusively demonstrate that SUB2 is essential for parasite viability, addressing a critical gap in our understanding of the parasite life cycle: how the host membrane is resealed following invasion.

**Decision letter after peer review:**

Thank you for submitting your article "The malaria parasite sheddase SUB2 governs host red blood cell membrane sealing at invasion" for consideration by *eLife*. Your article has been reviewed by three peer reviewers, one of whom is a member of our Board of Reviewing Editors, and the evaluation has been overseen by Dominique Soldati-Favre as the Senior Editor. The reviewers have opted to remain anonymous.

The reviewers have discussed the reviews with one another and the Reviewing Editor has drafted this decision to help you prepare a revised submission.

Summary:

The process of host cell invasion is an essential step in the life cycle of apicomplexan parasites, and involves many cell biological processes unique to these organisms. Central to these events is the interaction between parasite adhesins and the host cell membrane. The present study uses elegant genetics and a collection of analytical approaches to investigate the function of a parasite protease, SUB2, previously shown to mediate proteolysis of several key adhesins. The authors conclusively demonstrate that SUB2 is essential for parasite viability. Using conditional mutagenesis the authors provide precise insight into the timing and character of defects associated with SUB2 loss, including abortive invasion events that lead to host cell lysis or developmental arrest. Using mass spectrometry, the authors also confirm a role of SUB2 in shedding of MSP1, AMA1 and PTRAMP and uncover additional potential SUB2 substrates. By generating uncleavable MSP1 substrate they partially recapitulate the SUB2 phenotype, whilst the uncleavable form of AMA1 has no phenotype, suggesting that the lethal consequences of SUB2 disruption likely result from the combined effects of multiple substrates. They also present the phenotype of an AMA1 mutant using an efficient conditional strategy that shows its essential role for invasion. The present study addresses a critical gap in our understanding of apicomplexan invasion: how the host membrane is resealed following invasion. The experiments are accurately performed and well-presented but some conclusions require clarification and a more nuanced interpretation that acknowledges experimental limitations.

Essential revisions:

1) Differences between the mutants need to be more carefully considered. There is no evidence that the phenotype of SUB2 is linked to interference with AMA1 function, as proposed in the last paragraph of the Discussion. It is true that SUB2 and AMA1 mutants induce host cell lysis, but lysis seems to be the consequence of two different defects in the two mutants. In AMA1 mutants, that do not invade, the host cell lysis is likely due to injection of rhoptry proteins without stabilization of TJ, as proposed. By contrast, data suggest that lysis in SUB2 reflects a post-invasion failure, such as PVM scission defect and/or a host membrane repair defect. The narrative is too forceful in trying to reconcile what may be two separate phenotypes. The data on the AMA1 knockout could be removed without affecting the overall thrust of the story; alternatively, the results of the AMA1 knockout must be adequately contextualized. The authors should also more carefully discuss the differences between MSP1(PP) and SUB2 KO phenotypes noting that the delay in development appears very different (compare Figure 6D to Figure 4G). It is perhaps expected that a protease with many substrates would have pleiotropic effects that are not captured by individual substrates; however, formally speaking, the authors cannot rule out that there exists, among the other substrates of SUB2, one that captures the entire phenotype.

2) The data implicating SUB2 in membrane 'sealing' are indirect and alternative models should be discussed. Can the authors formally distinguish between adhesion, invasion, and membrane scission? For example, SUB2 knockout parasites may be hyper adhesive and enter cells through passive association rather than productive invasion. In other settings, such as sporozoite transmigration, parasites might be found inside of host cells without having secreted rhoptries, and such events would still be dependent on gliding motility, could elicit leakage and lysis of host cells, but would not appropriately establish a replicative niche. Have the authors examined rhoptry secretion in the context of SUB2 loss? If the SUB2 mutant is defective in sealing, one would expect the parasite to remain accessible to antibodies in the absence of RBC permeabilization. Albeit challenging, it might be worthwhile to determine whether surface proteins like AMA1 and MSP1 remain accessible to antibodies at the site of resealing in the absence of permeabilization (shortly after invasion). A time course experiment could help clarify whether lysis predominantly follows invasion, e.g. using phalloidin and Hoechst staining to enumerate infected and lysed cells after 15min, 1h, and 4h, similarly to what was done in Figure 5D. Incomplete shedding of MSP1 might also induce transient host cell damage that would affect the progression of growth, which could be tested by quantifying hemoglobin release and phalloidin staining. Excessive adhesion could create a physical obstacle to membrane resealing without SUB2 directly participating in the process. Similarly, host cell lysis could be a product of excessive traction on the membrane as the parasite moves against the host plasma membrane without releasing any points of contact-again explaining the observations without direct involvement of SUB2 in membrane resealing. The authors should consider the title of the manuscript in addressing these concerns and possibly modify their terminology (i.e. 'invasion defect' is used to describe the defect of the SUB2 when a defect in sealing but not internalization is proposed).

3) The authors conclude that Hb release does not occur in the presence of CytD, whilst there is a significant increase (p-value **). The same increase is observed in the SUB2 mutant. Is it linked to the addition of CytD on RBCs alone or a real effect of blocking parasite invasion at an early step? A control of RBCs with CytD without infection seems a necessary control for this experiment.

4) The mass spec experiment in Figure 3 is very interesting and should be analyzed in greater detail. E.g. what are proteins with a low score, there are two IMC associate proteins in this sample, is there an explanation? Naturally, there is much noise in mass spec so interpretations should be performed carefully. Figure 3 could be colored by all of the proteins expected to be secreted or on the surface, and additional controls (not expected to be shed). Figure 3—source data 1 should list razor and unique peptides separately, since several proteins are identified with a single peptide. Raw proteomics data should be deposited in an open archive.

5) The last paragraph of the Discussion proposes that the arrested post-invasion development of the SUB2 mutant is likely due to inhibition of DV biogenesis; however, there is no clear experimental evidence showing that MSP1-19 plays an active role in the biogenesis and function of the food vacuole during the intra-erythrocytic phase.

---

## [Author Response]

Essential revisions:1) Differences between the mutants need to be more carefully considered. There is no evidence that the phenotype of SUB2 is linked to interference with AMA1 function, as proposed in the last paragraph of the Discussion. It is true that SUB2 and AMA1 mutants induce host cell lysis, but lysis seems to be the consequence of two different defects in the two mutants. In AMA1 mutants, that do not invade, the host cell lysis is likely due to injection of rhoptry proteins without stabilization of TJ, as proposed. By contrast, data suggest that lysis in SUB2 reflects a post-invasion failure, such as PVM scission defect and/or a host membrane repair defect. The narrative is too forceful in trying to reconcile what may be two separate phenotypes. The data on the AMA1 knockout could be removed without affecting the overall thrust of the story; alternatively, the results of the AMA1 knockout must be adequately contextualized. The authors should also more carefully discuss the differences between MSP1(PP) and SUB2 KO phenotypes noting that the delay in development appears very different (compare Figure 6D to Figure 4G). It is perhaps expected that a protease with many substrates would have pleiotropic effects that are not captured by individual substrates; however, formally speaking, the authors cannot rule out that there exists, among the other substrates of SUB2, one that captures the entire phenotype.

The reviewer raises 3 separate points here: first, that the SUB2-null and AMA1-null mutants appear to induce RBC lysis in different ways; second, that there is a difference in severity between the intracellular death phenotype of the SUB2-null and uncleavable MSP1 mutants; and third that we cannot formally rule out the possibility that the entire SUB2-null phenotype could be due to the loss of cleavage of a single (possibly unknown) substrate. We accept all these points, and have now attempted to modify our revised text at the appropriate points in the manuscript in order to accommodate these concepts. We hope these changes are satisfactory.

2) The data implicating SUB2 in membrane 'sealing' are indirect and alternative models should be discussed. Can the authors formally distinguish between adhesion, invasion, and membrane scission? For example, SUB2 knockout parasites may be hyper adhesive and enter cells through passive association rather than productive invasion. In other settings, such as sporozoite transmigration, parasites might be found inside of host cells without having secreted rhoptries, and such events would still be dependent on gliding motility, could elicit leakage and lysis of host cells, but would not appropriately establish a replicative niche. Have the authors examined rhoptry secretion in the context of SUB2 loss? If the SUB2 mutant is defective in sealing, one would expect the parasite to remain accessible to antibodies in the absence of RBC permeabilization. Albeit challenging, it might be worthwhile to determine whether surface proteins like AMA1 and MSP1 remain accessible to antibodies at the site of resealing in the absence of permeabilization (shortly after invasion). A time course experiment could help clarify whether lysis predominantly follows invasion, e.g. using phalloidin and Hoechst staining to enumerate infected and lysed cells after 15min, 1h, and 4h, similarly to what was done in Figure 5D. Incomplete shedding of MSP1 might also induce transient host cell damage that would affect the progression of growth, which could be tested by quantifying hemoglobin release and phalloidin staining. Excessive adhesion could create a physical obstacle to membrane resealing without SUB2 directly participating in the process. Similarly, host cell lysis could be a product of excessive traction on the membrane as the parasite moves against the host plasma membrane without releasing any points of contact-again explaining the observations without direct involvement of SUB2 in membrane resealing. The authors should consider the title of the manuscript in addressing these concerns and possibly modify their terminology (i.e. 'invasion defect' is used to describe the defect of the SUB2 when a defect in sealing but not internalization is proposed).

Many conclusions regarding protein function reached on the basis of reverse genetic perturbations are inevitably ‘indirect’, in the sense that a direct mechanistic link between protein function and phenotype is often difficult to establish in a complex cellular setting. In this case, our conclusions implicating SUB2 activity in RBC membrane sealing are primarily based on: (1) the major phenotype (RBC lysis); and (2) our visual examination of the interactions between SUB2-null merozoites and target RBCs. We did our best to test the role of individual SUB2 substrates by the production of cleavage-resistance or null mutants but – as the reviewer points out above – our capacity to fully dissect the phenotype was ultimately hindered by the sheer range of different substrates affected by loss of SUB2. Given these experimental limitations, we acknowledge that there is a need to discuss alternative models, including effects on parasite adhesion. We have now included suitable text within the relevant sections of the revised Discussion. We maintain that the title of the manuscript is completely justified given the RBC lysis phenotype, which to our knowledge is unprecedented.

With regard to the specific experimental questions raised by the reviewer, we used target RBC echinocytosis as a proxy read-out for rhoptry discharge, since previous data from us and others (cited in the text) has convincingly linked rhoptry secretion to echinocytosis. We used EM to investigate the sealing defect, rather than investigating access of the intracellular parasite to antibodies. Regarding whether lysis can occur at extended time points following invasion by the ~50% of SUB2-null merozoites that could successfully establish rings, the time-course experiment suggested would likely not provide unambiguous results due to ongoing continued egress and invasion events in the cultures, which would continue anyway to increase the numbers of phalloidin positive cells in a time-dependent manner. However, In our own experiments, we found no evidence for further release of haemoglobin upon washing and reculturing cycle 1 rings produced by SUB2-null parasites. This suggests that following successful entry there was no further ‘leakage’ of haemoglobin. This is now referred to briefly in the relevant section of the Results.

3) The authors conclude that Hb release does not occur in the presence of CytD, while there is a significant increase (p-value **). The same increase is observed in the SUB2 mutant. Is it linked to the addition of CytD on RBCs alone or a real effect of blocking parasite invasion at an early step? A control of RBCs with CytD without infection seems a necessary control for this experiment.

This is a well-observed point, and refers to the data shown in Figure 5. We have now carried out the recommended control experiment, evaluating the effects of cytD or vehicle alone on Hb release from uninfected RBCs. Consistent with the data shown in Figure 5C, this showed that cytD treatment indeed induces a small but significant release of Hb from uninfected RBCs. Importantly, these findings do not alter our central conclusions from the experiments shown in Figure 5. Rather than modifying Figure 5. in the revised manuscript, the new control data are simply referred to in the revised Figure 5 legend and are presented as an additional supplementary figure (now Figure 5—figure supplement 1).

4) The mass spec experiment in Figure 3 is very interesting and should be analyzed in greater detail. E.g. what are proteins with a low score, there are two IMC associate proteins in this sample, is there an explanation? Naturally, there is much noise in mass spec so interpretations should be performed carefully. Figure 3 could be colored by all of the proteins expected to be secreted or on the surface, and additional controls (not expected to be shed). Figure 3—source data 1 should list razor and unique peptides separately, since several proteins are identified with a single peptide. Raw proteomics data should be deposited in an open archive.

For the original submission we felt that, within the context of the main thrust of the paper, an extended discussion of the mass spec data would detract from the flow of the text. However we fully agree with the reviewer that the mass spec data are very interesting and warrant additional discussion. So for our revised submission we have followed the reviewers’ recommendations, expanding the relevant section of the Discussion section. We have also colour-coded Figure 3, modified Figure 3—source data 1 as suggested, and deposited the raw proteomic data with the ProteomeXchange Consortium via the PRIDE partner repository.

5) The last paragraph of the Discussion proposes that the arrested post-invasion development of the SUB2 mutant is likely due to inhibition of DV biogenesis; however, there is no clear experimental evidence showing that MSP1-19 plays an active role in the biogenesis and function of the food vacuole during the intra-erythrocytic phase.

We entirely accept that there is no experimental evidence implicating MSP1-19 in DV biogenesis; the available evidence (see Dluzewski et al., 2008) simply shows that MSP1-19 is the first known marker for the developing DV membrane, indicating that plasma membrane constituents of the newly-invaded parasite are involved in biogenesis of the DV membrane. However, our speculative suggestion to explain the phenotype of the uncleavable MSP1 mutants is not that MSP1-19 is required for DV formation, but that a lack of MSP1 shedding interferes with DV biogenesis, due to the bulky surface coat components remaining on the plasma membrane of the intracellular merozoite. The distinction is subtle but key, and we believe this speculation is completely justified. Given this, we have now slightly modified and expanded our text in the earlier part of the Discussion to clarify our thinking on this point. We hope this is acceptable.